# R-loop proximity proteomics identifies a role of DDX41 in transcription-associated genomic instability

Thorsten Mosler[1], Francesca Conte[1], Gabriel M. C. Longo [1], Ivan Mikicic [1], Nastasja Kreim[1], Martin M. Möckel[1], Giuseppe Petrosino[1], Johanna Flach[2], Joan Barau [1], Brian Luke [1,3], Vassilis Roukos [1] & Petra Beli [1,3 ✉]

Transcription poses a threat to genomic stability through the formation of R-loops that can obstruct progression of replication forks. R-loops are three-stranded nucleic acid structures formed by an RNA–DNA hybrid with a displaced non-template DNA strand. We developed RNA–DNA Proximity Proteomics to map the R-loop proximal proteome of human cells using quantitative mass spectrometry. We implicate different cellular proteins in R-loop regulation and identify a role of the tumor suppressor DDX41 in opposing R-loop and double strand DNA break accumulation in promoters. DDX41 is enriched in promoter regions in vivo, and can unwind RNA–DNA hybrids in vitro. R-loop accumulation upon loss of DDX41 is accompanied with replication stress, an increase in the formation of double strand DNA breaks and transcriptome changes associated with the inflammatory response. Germline loss-of-function mutations in *DDX41* lead to predisposition to acute myeloid leukemia in adulthood. We propose that R-loop accumulation and genomic instability-associated inflammatory response may contribute to the development of familial AML with mutated DDX41.

[1] Institute of Molecular Biology (IMB), Mainz, Germany. [2] Department of Hematology and Oncology, Medical Faculty Mannheim of the Heidelberg University, Mannheim, Germany. [3] Institute of Developmental Biology and Neurobiology (IDN), Johannes Gutenberg-Universität, Mainz, Germany. ✉email: p.beli@imb-mainz.de

Transcription by RNA polymerase II (RNAPII) is essential to all cellular processes and hence to the adaptive response of cells to internal and external stimuli. Dysregulated transcription resulting in high transcription rates and increased frequency of transcription–replication conflicts is observed in many tumors. Accordingly, targeting different mechanisms that enable tumor cells to cope with transcription stress is being explored as a therapeutic strategy[1]. Co-transcriptional R-loops are three-stranded nucleic acid structures formed by an RNA–DNA hybrid with a displaced non-template DNA strand. R-loops are most prevalent at gene promoters where they regulate transcription initiation[2,3]. R-loop formation at CpG islands (CGIs) that are present at ~60% of gene promoters protects these regions from DNA methylation by repelling DNA methyl-transferases (DNMTs)[2,4]. In addition to reducing the binding of DNMTs, R-loop-dependent recruitment of GADD45A and active de-methylation by TET enzymes has been proposed as mechanism for loss of CGIs methylation[3]. Furthermore, R-loops are present at the 3′ end of genes where they facilitate transcription termination by stalling RNAPII downstream of the poly-adenylation sequence[5,6]. R-loops are prevalent in highly transcribed genes and accumulate in repeats such as centromeres, telomeres, and retrotransposons[7–11]. Genome-wide approaches for mapping R-loops in human cells revealed that R-loops occupy up to 5% of unique sequences[12]. The presence of G-quadruplexes (G4s) on the displaced DNA contributes to the stabilization of R-loops[13]. Modification of RNA in the hybrid with N6-methyladenosine (m6A) provides an additional layer of regulation through the recruitment of m6a reader proteins[14–16]. In addition to the regulatory functions of R-loops in transcription, DNA repair, telomere maintenance, and chromosome segregation, these non-B DNA structures can be drivers of genomic instability[17–22]. The single-stranded DNA in the R-loops is more prone to DNA damage[23,24]. Stalled transcription complexes at R-loops can trigger their processing by nucleotide excision repair endonucleases ERCC4 (also known as XPF) and ERCC5 (also known as XPG) into double-strand breaks (DSBs)[19]. In cycling cells, R-loops can be formed as a consequence of head-on transcription–replication conflicts[25–27].

Different proteins regulate R-loop levels in human cells either by preventing their formation or by assisting their resolution. RNA-binding proteins that bind to nascent RNAs and are involved in the maturation or export of mRNA, such as the THO complex or the nuclear exosome, oppose R-loop formation[27,28]. Furthermore, negative supercoiling of DNA that is normally relaxed by Topoisomerase 1 (TOP1) favors R-loop formation[25,29]. Once formed, R-loops can be removed by the action of Ribonuclease H1 (RNaseH1) and H2 (RNaseH2)—conserved endonucleases that hydrolyze the phosphodiester backbone of the RNA moiety in RNA–DNA hybrids[30]. RNaseH1 acts as a monomer and harbors an N-terminal hybrid-binding domain (HBD) and a C-terminal catalytic domain. In vitro, the HBD (residues 27–76) displays at least 25-fold higher affinity for RNA–DNA hybrids as compared to dsRNA[31,32]. Recombinant GFP-tagged, catalytically inactive (D210N) human RNaseH1 was recently reported as a sensitive and specific tool for in situ imaging of RNA–DNA hybrids in fixed cells[33]. In addition to RNase H enzymes, different helicases including SETX, DDX5, and DDX39B have also been implicated in the unwinding of RNA–DNA hybrids and resolution of R-loops[5,18,34].

Dead box helicase 41 (DDX41) is a tumor suppressor that is conserved in *D. melanogaster*, *C. elegans*, *D. rerio*, and plants, and is considered essential for cell growth and viability[35–38]. Somatic and germline mutations in *DDX41* are present in 0.5 to 4% of adult myelodysplastic syndrome (MDS)/acute myeloid leukemia (AML) cohorts and are considered as oncogenic drivers[39].

Pathogenic germline variants in *DDX41* predominantly lead to frameshifts and production of truncated protein forms, whereas somatic mutations are mostly located within the DEAD box and helicase domain likely resulting in compromised helicase activity[40]. DDX41 has been reported to interact with components of the spliceosome, and *DDX41* deletion or mutations led to splicing defects and faulty RNA processing[41]. The role of DDX41 in RNA processing appears to be conserved as the *C. elegans* orthologue, SACY-1, was recently shown to associate with the spliceosome, and to impact the transcriptome through splicing-dependent and -independent mechanisms[36]. Despite the relevance of DDX41 in cancer, the cellular and molecular functions of DDX41 remain poorly understood.

We employed quantitative mass spectrometry (MS)-based proteomics to identify proteins that regulate R-loops in human cells. To this end, we developed RNA–DNA Proximity Proteomics (RDProx) that enables mapping of the R-loop-proximal proteome using the fusion protein of the hybrid-binding domain (HBD) of RNaseH1 and an engineered variant of ascorbate peroxidase (APEX2). We implicated proteins with different cellular functions in R-loop regulation and characterized the role of the tumor suppressor DDX41 in opposing R-loops and DSBs in promoters. We demonstrate that DDX41 preferentially associates with promoter regions and that recombinant DDX41 can bind and unwind RNA–DNA hybrids in vitro. We propose that the accumulation of co-transcriptional R-loops, and consequently replication stress, DSBs, and an inflammatory response may collectively contribute to the development of familial AML and MDS with mutated DDX41.

## Results

**RDProx identifies the R-loop proximal proteome.** Tight regulation of R-loop levels across the genome is essential for their function in promoting chromatin-associated processes and for preventing R-loop-dependent genomic instability. To gain insights into protein-based mechanisms that regulate R-loop homeostasis, we probed the R-loop-proximal protein networks using RNA–DNA proximity proteomics (RDProx). We fused the HBD (residues 27–76) of RNaseH1 to an engineered variant of soybean ascorbate peroxidase (APEX2)[42]. As a negative control, we employed a construct harboring three point mutations in the HBD (HBD-WKK) that led to a loss of affinity towards RNA–DNA hybrids (Supplementary Fig. 1a)[31]. In accordance, only GFP-tagged HBD and not the WKK mutant associated with chromatin under pre-extraction conditions (Supplementary Fig. 1b). The preferential binding behavior of the HBD for RNA–DNA hybrids was confirmed in vitro using purified domains and different nucleic acid substrates. As expected, the WKK mutant displayed dramatically reduced affinity for hybrids (Supplementary Fig. 1c, d). APEX2-HBD or APEX2-HBD-WKK fusion proteins were expressed in HEK293T cells and labeling of proximal proteins was induced in vivo (Fig. 1a and Supplementary Fig. 1e). Biotinylated proteins were enriched using streptavidin and analyzed by liquid chromatography (LC)-tandem mass spectrometry (MS/MS). Stable isotope labeling with amino acids in cell culture (SILAC) was used to distinguish proteins that are proximal to the wild-type HBD compared to the WKK mutant (Fig. 1a). We performed three replicate experiments that showed excellent reproducibility ($r > 0.85$) and identified 312 proteins enriched with high confidence by RDProx ($\log_2$ FC > 2; FDR < 0.01) (Fig. 1b and Supplementary Fig. 1f, Supplementary Data 1). Among these proteins, we identified previously known R-loop regulators such as TOP1, AQR, single-stranded DNA-binding proteins RPA1/2, RNaseH2A, components of the THO complex (THOC1/2/6, THOC6, ALYREF), and the nuclear exosome

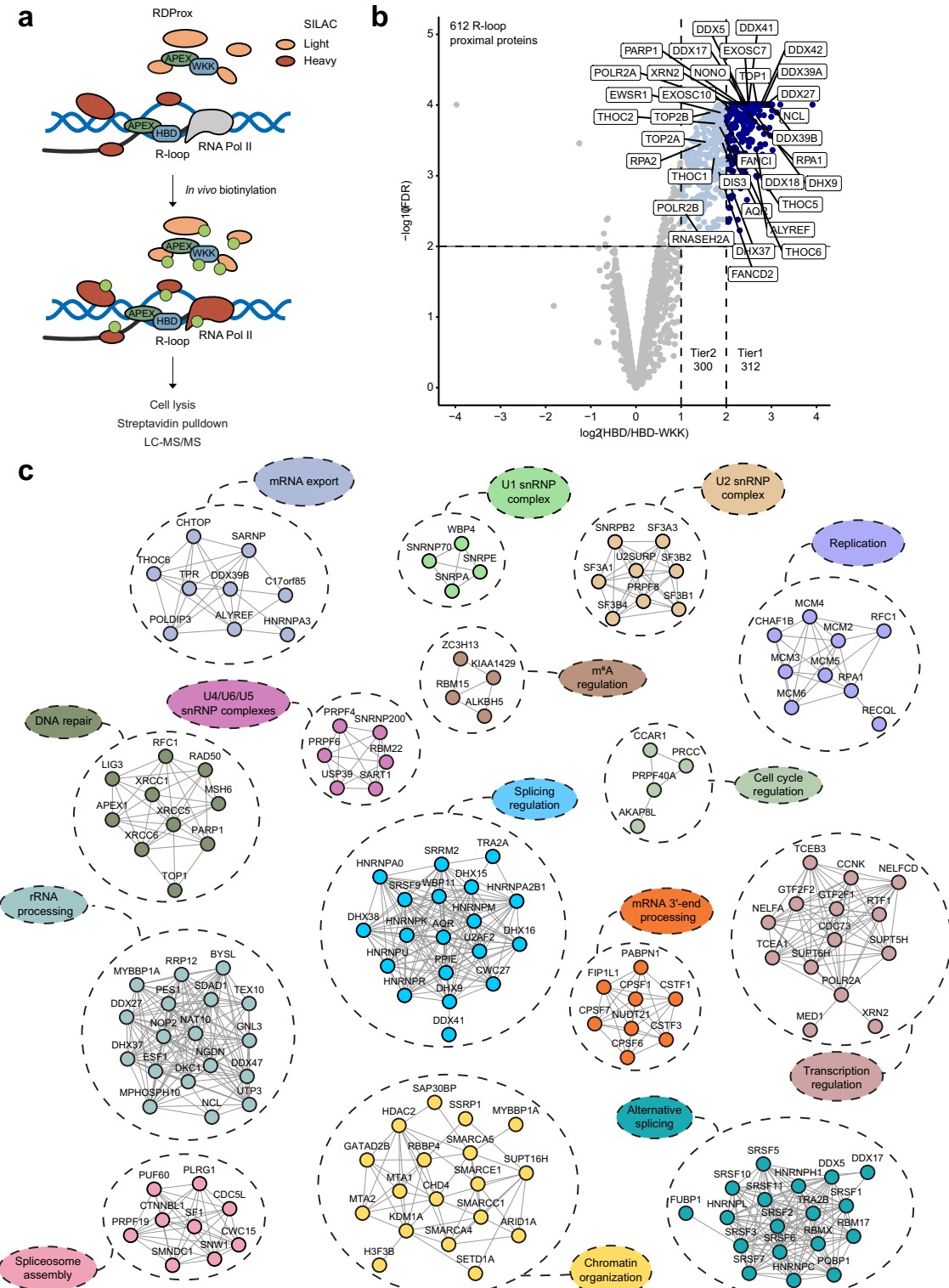

**Fig. 1 RDProx-Mapping R-loop-proximal proteome on native chromatin. a** Schematic representation of the RDProx workflow for identification of R-loop-proximal proteins. HBD or HBD-WKK fused N-terminally to APEX2 were transiently expressed in light or heavy SILAC-labeled HEK293T cells. Biotinylation was induced upon the addition of 500 μM biotin–phenol for 2 h at 37 °C and 1 mM $H_2O_2$ for 2 min at room temperature. Samples were pooled after cell lysis and biotinylated proteins purified using NeutrAvidin beads. Denatured proteins were separated by SDS-PAGE and in-gel digested before LC-MS/MS analysis. **b** Volcano plot of protein groups identified by RDProx in $n = 3$ biologically independent experiments. Mean $\log_2$ ratios of all replicates between HBD and HBD-WKK are plotted against the $-\log_{10}$ FDR. The FDR and enrichment were calculated using Limma[103]. Significantly enriched proteins are highlighted in blue (FDR < 0.01). Light blue indicates proteins in Tier 2 (300 proteins) above 2-fold change of the mean ratio and dark blue indicates proteins in Tier 1 (312 proteins) with a 4-fold change or higher. **c** Functional interaction network of proteins identified by RDProx. Genes were manually annotated based on literature and corresponding GO terms (Biological Process and Molecular Function). Clusters were generated based on the manual annotation. Edges between the nodes indicate interactions based on STRING with a confidence score equal or above 0.7.

(EXOSC7, EXOSC10). R-loop proximal proteins showed functional interactions as demonstrated by the identification of different protein clusters involved in splicing, m6A regulation, mRNA 3′ end processing, mRNA export, transcription regulation, chromatin organization, and DNA replication/repair (Fig. 1c and Supplementary Fig. 1h). These proteins were enriched with domains typical for RNA- and DNA-binding proteins including RRM, helicase, DEAD/DEAH, CID domain, MCM N-terminal domain, RNA polymerase II binding domain, SAP domain, MCM OB domain, and K Homology domain (Supplementary Fig. 1g).

**DDX41 loss leads to replication stress and R-loop-dependent genomic instability.** To assess the possible function of the identified DEAD box helicases in the regulation of R-loops, we monitored the intensity of Ser139 phosphorylation on the histone variant H2AX (γH2AX; proxy for DSBs) upon depletion of DDX27, DDX41, DDX42, DHX37, and DDX39A. Only depletion of DDX41 and AQR led to a notable increase in nuclear γH2AX intensity in unchallenged conditions (Fig. 2a). Dead box helicase 41 (DDX41) is a poorly characterized tumor suppressor and pathogenic variants in DDX41 cause familial MDS and AML, which prompted us to investigate its potential role in R-loop metabolism[41]. We confirmed that the proximity of DDX41 to R-loops is reduced when R-loops are suppressed by overexpression of RNaseH1 under the control of a doxycycline-inducible promoter (Supplementary Fig. 2a). Increased levels of γH2AX in DDX41 knockdown cells were confirmed by western blotting (Supplementary Fig. 2b, c). Overexpression of RNaseH1 partially rescued the effect of DDX41 knockdown on γH2AX pointing to R-loop-dependent genomic instability (Fig. 2b). Overexpression of RNaseH1 also rescued the increased formation of DSBs in DDX41 knockdown cells measured by neutral comet assay (Fig. 2c). Increased formation of DSBs in DDX41 knockdown cells was confirmed by monitoring TP53BP1 (also known as 53BP1) foci formation (Fig. 2d, Supplementary Fig. 2d). Knockdown of DDX41 resulted in increased phosphorylation of RPA on Ser33 and significantly reduced DNA fiber length similarly to mild replication stress induced by DNA polymerase inhibition with 100 nM aphidicolin (Fig. 2e, f). Accordingly, an increase in γH2AX and pRPA intensity was predominant in the S phase (Supplementary Fig. 2e, f). Moreover, the phosphorylation of RPA was, to some extent, mediated by the replication stress kinase ATR, since ATR inhibition with VE-821 significantly reduced the increase in pRPA intensity after DDX41 knockdown (Supplementary Fig. 2g). To corroborate that these cells depend on ATR activity to respond to replication stress, we treated U2OS cells after DDX41 knockdown, and OCI-AML3 cells expressing DDX41 disease variants (L237F/P238T and R525H), with ATR inhibitors and monitored their viability. Both DDX41 knockdown and disease variants-expressing cells displayed sensitivity to ATR inhibition suggesting that AML cells with pathogenic DDX41 variants also display replication stress (Supplementary Fig. 2h, i).

**DDX41 unwinds RNA–DNA hybrids in vitro and its loss results in R-loop accumulation.** We confirmed that DDX41 is proximal to R-loops in human cells using proximity ligation assays with antibodies against endogenous DDX41 and GFP-tagged HBD or HBD-WKK (Fig. 3a and Supplementary Fig. 3a). The proximity to R-loops and the occurrence of spontaneous DNA damage and replication stress in DDX41 knockdown cells, prompted us to investigate whether DDX41 opposes the accumulation of R-loops. To test this, we performed dot-blot analysis with the S9.6 RNA–DNA hybrid antibody. Indeed, knockdown of DDX41 resulted in the accumulation of RNA–DNA hybrids that were sensitive to RNaseH1 overexpression (Fig. 3b). Using

chromatin-bound GFP-tagged HBD as a proxy for R-loops, we could confirm increased levels of R-loops upon depletion of DDX41 and AQR as a positive control (Fig. 3c and Supplementary Fig. 3b, c). Inhibition of transcription by treating cells with DRB partially rescued the effect of DDX41 knockdown on R-loop accumulation (Fig. 3c and Supplementary Fig. 3b). As a consequence of inhibiting transcription elongation, DRB also induces R-loop formation, which could explain the incomplete rescue of DRB on R-loop accumulation in DDX41 knockdown cells[43].

Our experiments indicated that DDX41 opposes the accumulation of R-loops and R-loop-dependent genomic instability. To address whether DDX41 can directly bind and unwind RNA–DNA hybrids in R-loops, we purified full-length DDX41, DDX41 lacking the helicase domain (153–410), and the AML-associated R525H variant in the C-terminus of the RecA-like helicase core domain (Supplementary Fig. 3d, e). Full-length DDX41 was incubated with five different fluorophore-conjugated oligonucleotide substrates to determine the binding affinity. Binding of DDX41 to the substrates resulted in a change of fluorescence polarization. We found that recombinant DDX41 possesses the strongest affinity ($K_d = 2.5\ \mu M \pm 1.4\ \mu M$) for RNA–DNA hybrids in vitro compared to other nucleic acid substrates (Fig. 3d). The lack of known $K_d$ values for other RNA–DNA helicases make it difficult to draw comparisons, but we note that HBD's affinity for RNA–DNA hybrids is ~10× higher ($K_d = 190\ nM \pm 30$) (Supplementary Fig. 1c). Furthermore, we employed an ATPase assay to determine whether the ATPase domain of DDX41 hydrolyzes ATP when encountering an RNA–DNA hybrid substrate. We found that that ATP hydrolysis by DDX41 was stimulated by RNA–DNA hybrids with a single-stranded DNA overhang (Supplementary Fig. 3f). To test whether the ATPase activity was accompanied by the unwinding of this substrate, we established a FRET-based displacement assay. The separation of a fluorophore-labeled DNA strand and a quencher-conjugated RNA strand resulted in increased fluorescence intensity. Importantly, DDX41 not only bound but also unwound the RNA–DNA hybrids in vitro in a concentration-dependent manner, whereas DDX41 lacking the helicase domain was not able to separate the two strands (Fig. 3e, f). Also, DDX41 harboring the disease-associated R525H variant showed decreased efficiency in RNA–DNA hybrid unwinding (Fig. 3f and Supplementary Fig. 3h). Taken together, recombinant full-length DDX41 preferentially binds RNA–DNA hybrids compared to other nucleic acids and can unwind RNA–DNA hybrids in vitro.

**DDX41 opposes R-loop accumulation in promoters.** To test whether DDX41 can associate with chromatin in vivo, we generated U2OS cells that express GFP-tagged DDX41 under a doxycycline-inducible promoter. We confirmed that these cells show pan-nuclear DDX41 staining that mirrored the localization of endogenous DDX41 (Supplementary Fig. 4a). We used a GFP nanobody to target micrococcal nuclease to chromatin regions bound by DDX41 using greenCUT&RUN[44,45]. We detected 19,327 DDX41 peaks in 2 biologically independent experiments, among which 6,363 were consistently found in both experiments (Supplementary Data 2). Interestingly, DDX41 displayed a preference to bind promoters with 41% of DDX41 peaks mapping to promoter regions (TSS ± 3 kb) (Fig. 4a, b). We identified 6,441 promoters bound by DDX41, and a comparison of DDX41 binding sites with RNA-sequencing from U2OS cells revealed that DDX41 association with promoters depends on gene expression levels with the association being stronger at highly expressed genes (Supplementary Fig. 4b).

To quantify R-loops genome-wide in wild-type U2OS cells and upon knockdown of DDX41, we performed MapR that uses a

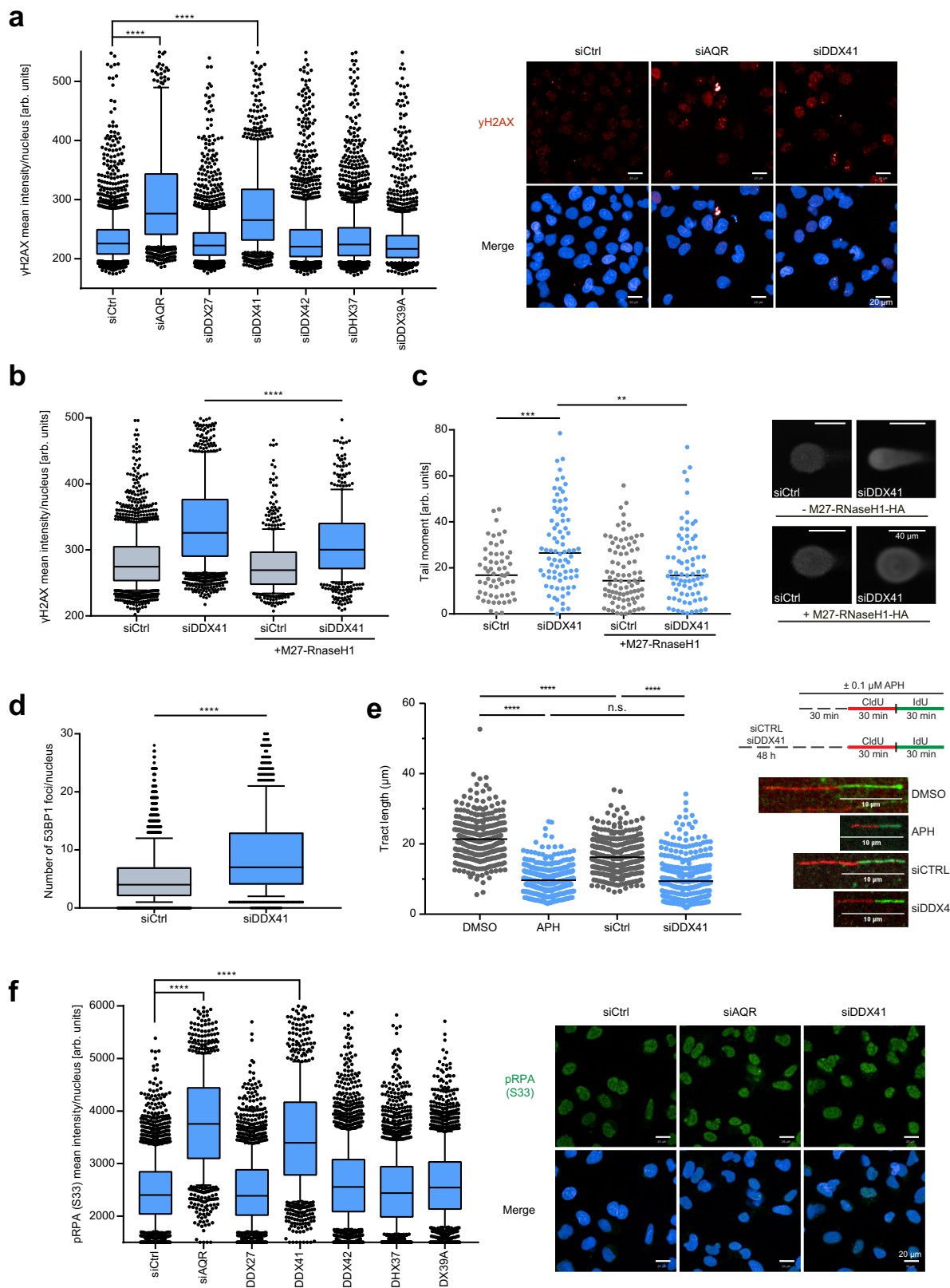

catalytically-dead *E. coli* Ribonuclease H to target micrococcal nuclease to R-loops, which are subsequently cleaved, released, and identified by sequencing[46,47]. 47% of R-loops were identified in promoter regions (TSS ± 3 kb) and the levels of R-loops in promoters positively correlated with gene expression (Fig. 4c and Supplementary Fig. 4c, d, Supplementary Data 3, 4). Inhibition of

transcription by Actinomycin D resulted in a dramatic decrease of R-loops in promoters (Fig. 4c and Supplementary Fig. 4d). Importantly, knockdown of DDX41 led to a significant increase of R-loops in promoters (Fig. 4c and Supplementary Fig. 4d, e). This was also reflected by the increased number of MapR peaks detected in DDX41 knockdown cells in comparison to wild-type

**Fig. 2 DDX41 depletion leads to replication stress and genomic instability. a** Immunofluorescence analysis of γH2AX in U2OS cells 48 h after indicated knockdowns. Center lines of boxplots indicate the median, the limits the 25th–75th percentile, whiskers the 10th–90th percentile, dots outliers. Representative data of $n = 3$ biologically independent experiments; p-values ($p < 0.0001$, $p = 0.1788$, $p < 0.0001$, $p = 0.8259$, $p > 0.9999$, $p = 0.9569$) were derived from >1000 cells using one-way ANOVA with Tukey correction for multiple comparisons. Representative images of γH2AX (red) staining and Hoechst33342 (blue). Scale bars—20 µm. **b** Immunofluorescence analysis of γH2AX in U2OS cells ± doxycycline-inducible GFP-tagged M27-RNaseH1. Quantification of cells with medium GFP intensity (medium M27-RNaseH1 expression). Representative boxplots of $n = 2$ biologically independent experiments. Center of boxplots indicates the median, limits the 25th–75th percentile, whiskers the 10th–90th percentile, dots outliers. $p < 0.0001$ derived from $n > 500$ cells using a two-sided Mann–Whitney test. **c** Single-cell electrophoresis of U2OS cells 48 h after knockdown ± doxycycline-inducible expression of HA-tagged M27-RNaseH1. Representative images are displayed (right). Scale bars—40 µm. Dots depict individual tail moments, black line the median. Representative results from $n = 2$ biologically independent experiments. p-values ($p = 0.0001$, $p = 0.0031$) were derived from $n > 50$ cells using one-way ANOVA with Tukey correction for multiple comparisons. **d** Immunofluorescence analysis of 53BP1 foci in U2OS cells 48 h after indicated knockdowns. Whiskers of the box plot represent the 10th–90th percentile, the center line the median, the limits the 25th–75th percentile, and the dots depict outliers. Representative of $n = 3$ biologically independent experiments. p-value < 0.0001 was derived from $n > 1000$ cells using an unpaired, two-sided Student's t-test. **e** DNA fiber spreading assay of U2OS cells after 48 h knockdown of DDX41. Controls were either treated with DMSO or 100 nM aphidicolin for 1.5 h. Representative images (white line indicates 10 µm scale) and quantifications of fiber tract length. Dots represent individual values and the black line the median. At least 260 fibers were quantified across $n = 1$ experiment. p-values ($p < 0.0001$, $p < 0.0001$, $p < 0.0001$, $p = 0.5794$) were derived using one-way ANOVA with Tukey correction for multiple comparisons. **f** Immunofluorescence analysis of pRPA (Ser33) in U2OS cells 48 h after indicated knockdowns. Representative images (right): Hoechst33342 (blue), pRPA (Ser33) (green). Center of boxplots indicates the median, limits the 25th–75th percentile, whiskers the 10th–90th percentile, dots outliers. Representative data of $n = 3$ biologically independent experiments are displayed. p-values ($p < 0.0001$, $p > 0.9999$, $p < 0.0001$, $p < 0.0001$, $p = 0.9384$, $p < 0.0001$) were derived from $n > 1000$ cells using one-way ANOVA with Tukey correction for multiple comparisons. Scale bars—20 µm. Source data are provided as a Source Data file.

cells: In wild-type U2OS cells we detected 24,492, whereas in DDX41 knockdown cells 35,627 (Supplementary Data 3). We identified changes (FC > 2) in the MapR signal in 7,315 genomic regions in the absence of DDX41 (Fig. 4d). 6,810 regions displayed an increase in the MapR signal (i.e., R-loop gain) after DDX41 knockdown, of which 81% overlapped with promoter regions and 74% overlapped with CGI promoters (Fig. 4e, f). In contrast, none of the 505 regions with decreased MapR signal (i.e., R-loop loss) mapped to CGI promoters (Fig. 4f). Regions with R-loop loss overlapped with introns and distal intergenic regions (Fig. 4e). 5,506 (29%) out of 18,811 of R-loops in promoter regions showed a gain (FC > 2) in DDX41 knockdown cells, pointing to a prominent role of DDX41 in opposing R-loops in promoters of active genes (Supplementary Data 3). Reactome pathway over-representation analysis of genes with accumulated R-loops revealed chromatin organization, NOTCH, and TGFβ signaling (Fig. 4g). Nearly 40% of promoters that displayed accumulation of R-loops also associated with GFP-tagged DDX41 (Fig. 4h and Supplementary Fig. 4f). This is likely an under-estimation since CUT&RUN was performed under very mild crosslinking conditions, and DDX41 loosely associates with chromatin. Accumulation of R-loops at promoter regions was not accompanied by a global defect in nascent transcription based on 5-Ethynyl Uridine (EU) incorporation (Supplementary Fig. 4g). However, we observed a mild but significant decrease in serine 5- and an increase in serine 2-phosphorylation of the RBP1 C-terminal domain (CTD), suggesting that RNAPII initiation and possibly elongation are perturbed by accumulated R-loops in promoter regions (Supplementary Fig. 4h).

**DDX41 loss increases DNA fragility in promoters and induces inflammatory response.** To investigate whether the genomic instability observed upon loss of DDX41 derives from DSBs and whether the sites of DSBs coincide with sites of R-loop accumulation, we performed sBLISS (Break Labeling In Situ and Sequencing) in wild type and DDX41 knockdown HCT116 cells[48,49]. In accordance with previous studies[49–52], the majority of endogenous DNA fragility hotspots in unchallenged cells were mapped to promoters and this phenotype was even more pronounced in DDX41 knockdown cells where 63% or 5,307 out of 8,381 of DNA fragility hotspots were mapped to promoters (Fig. 5a–c). In total 8,381 DNA fragility hotspots were identified

in DDX41 knockdown cells, of which 5,958 were not present in wild-type HCT116 cells (Fig. 5d and Supplementary Data 5). We found 3,108 DSB gains (FC > 2) in DDX41 KD cells, 54% of which mapped to promoter regions (Supplementary Fig. 5a and Supplementary Data 5). To investigate whether and to which extent R-loops correlate with DSBs and whether DSBs in DDX41 knockdown cells overlap with promoters displaying R-accumulation, we performed MapR in HCT116 cells. We compared the MapR with previously published GRO-sequencing data in HCT116 cells to test the dependency of R-loops on transcription[53]. Similar to U2OS cells, loss of DDX41 in HCT116 cells led to a dramatic accumulation of R-loops in promoters of active genes (Supplementary Fig. 5b, c). From 15,177 MapR peaks identified in all 3 replicate experiments in DDX41 knockdown cells, 7,275 showed a gain in R-loops (FC > 1.5) (Supplementary Fig. 5d and Supplementary Data 6). Using sBLISS, we identified 1,642 promoter regions that showed increased fragility in DDX41 knockdown cells, and 53% of those promoters with DSB gains also displayed R-loops upon DDX41 loss (Fig. 5e, f and Supplementary Fig. 5e). This suggests that a large proportion of DNA fragile sites in promoters coincides with R-loops in DDX41 knockdown cells.

Interestingly, RNA-sequencing revealed upregulation of genes involved in inflammatory signaling in DDX41 knockdown cells, in particular NF-kB signaling, which was confirmed also by increased nuclear localization of NF-kB subunit p65 in these cells (Fig. 5h and Supplementary Fig. 5f). Inflammatory signaling genes did not display an accumulation of R-loops nor were bound by DDX41, suggesting that these genes are not directly regulated by DDX41, but are likely an indirect consequence of DDX41-dependent R-loop accumulation, and genomic instability. To test whether DDX41 also opposes DSBs and genomic instability in hematopoietic stem and progenitor cells (HSPCs) in which loss-of-function mutations of DDX41 result in AML, we depleted DDX41 in human CD34+ HSPCs, and monitored DSBs using 53BP1 foci. Importantly, we found that depletion of DDX41 using two different shRNAs led to spontaneous DSB formation (Fig. 6a). Notably, overexpression of DDX41 variants found in AML patients in either the DEAD (L237F/P238T) or helicase domain (R525H) resulted in the accumulation of 53BP1 foci in CD34+ cells (Fig. 6b). These results suggest that also in human HSPCs DDX41 functions in opposing transcription-associated genomic instability.

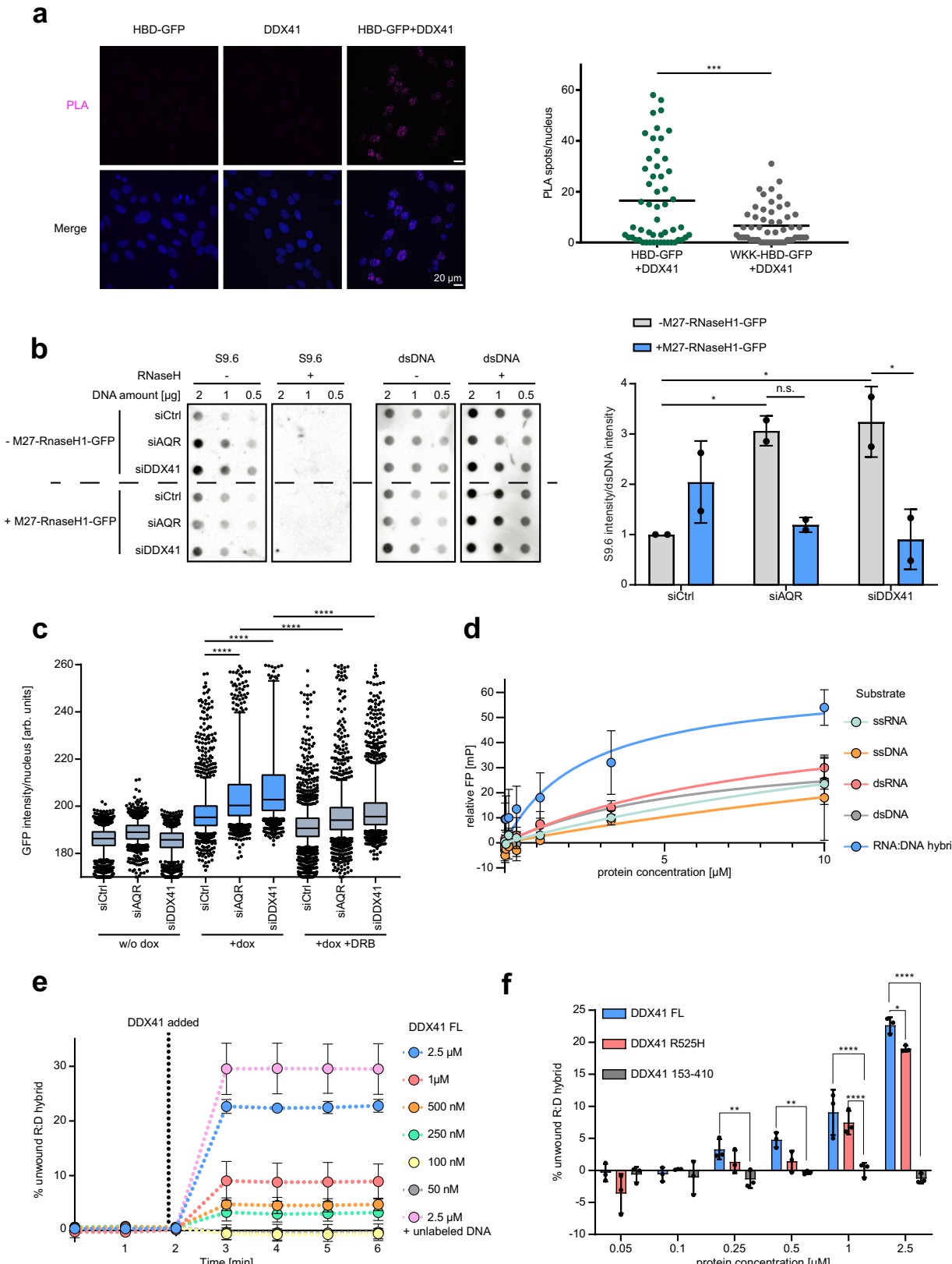

## Discussion

R-loop levels across the genome need to be balanced to ensure the regulation of chromatin-associated processes without inflicting DNA damage and genomic instability. Here, we developed RDProx that provides a snapshot of the R-loop-proximal proteome in human cells. We identified 612 R-loop-proximal proteins and

divided them in two categories (Tier 1 and Tier 2) depending on the probability of their presence at R-loops, providing a rich resource for further functional investigations. The advantages of RDProx are manifold: (1) labeling of R-loop-proximal proteins is performed in vivo, which ensures that R-loops, chromatin, and cellular compartments remain intact; (2) chromatin-associated proteins that are

**Fig. 3 DDX41 opposes R-loop accumulation and can unwind hybrids in vitro. a** Proximity ligation assay (PLA) between endogenous DDX41 and GFP-tagged HBD or HBD-WKK. Representative images and quantification of nuclear PLA spots. Dots represent results from individual cells, black line indicates the median. $p$-value < 0.0001 derived from > 50 cells from $n = 1$ experiment using two-sided Student's $t$-test. Scale bars—20 μm. **b** S9.6 and double-stranded DNA dot-blot analysis of U2OS cells expressing GFP-tagged M27-RNaseH1 upon doxycycline (dox) after 48 h of indicated knockdowns. Representative images (left). Data are represented as mean ± standard deviation (right). Black dots represent individual results from $n = 2$ biologically independent experiments. $p$-values ($p = 0.437$, $p = 0.653$, $p = 0.338$, $p = 0.0281$) derived using one-way ANOVA with Tukey correction for multiple comparisons. **c** HBD-GFP retention assay after indicated 48 h knockdowns in U2OS cells. Control cells were treated with 100 μM DRB for 3 h. Center of boxplots indicates the median of the population, limits the 25th–75th percentile, whiskers the 10th–90th percentile, and dots represent outliers. Representative data of $n = 3$ biologically independent experiments are shown. $p$-values ($p < 0.0001$, $p < 0.0001$, $p < 0.0001$, $p < 0.0001$) derived from $n > 1000$ cells using one-way ANOVA with Tukey correction for multiple comparisons. **d** Fluorescence polarization (FP) assay of full-length DDX41 and indicated 6-FAM-conjugated oligonucleotides in $n = 2$ independent experiments with individually thawed protein aliquots. The protein concentration on a $\log_2$-scale is plotted against the FP in mP (milipolarization unit). Data are represented as mean values ± standard deviation. Colored lines represent Michaelis–Menten fits. **e** FRET-based RNA–DNA hybrid displacement assay. Titrated full-length (FL) DDX41 is incubated with 100 nM RNA–DNA hybrid substrate and 5 μM ATP. Displacement of the IBFQ-conjugated 38-mer DNA oligo from the 6-FAM-conjugated 13-mer RNA oligo was measured by the change in fluorescence intensity. Data of $n = 3$ independent experiments with individually thawed proteins are represented as mean values ± standard deviation ($n = 2$ for 2.5 μM, unlabeled DNA). **f** Displacement assay from **e** using titrated full-length (FL) DDX41, R525H, or 153–410 mutant. Data are represented as mean ± standard deviation. Dots indicate results of $n = 3$ independent experiments with individually thawed proteins. $p$-values ($p = 0.005$, $p = 0.0021$, $p < 0.0001$, $p = 0.0329$, $p < 0.0001$) derived by two-way ANOVA with Tukey correction for multiple comparisons. Source data are provided as a Source Data file.

difficult to solubilize are amenable to the analysis, and (3) low affinity and transient interactions are detected. RDProx is easily applicable for mapping R-loop-proximal proteins in different cell lines and species as well as upon different cellular perturbations. Thereby, it provides a methodological framework to answer outstanding questions, including how R-loop regulation differs between cell cycle stages or in response to stress that impacts transcription or co-transcriptional processes. Previous studies have employed the S9.6 RNA–DNA hybrid antibody for immunoprecipitation of proteins that associate with R-loops or used an in vitro-generated RNA–DNA hybrid to pull down interacting proteins from cell extracts[54,55]. Recent reports showed that the S9.6 antibody in fixed human cells predominantly recognizes ribosomal RNA and not RNA–DNA hybrids[56]. Unbiased inspection of the previously reported S9.6-based proteomics data set by GO term enrichment analysis revealed "rRNA processing" and "ribosome biogenesis" as the most significantly enriched terms (Supplementary Fig. 1f). RDProx relies on the HBD of RNaseH1 and is therefore inherently biased to the R-loops that are recognized and bound by RNaseH1. Recently, the existence of different classes of R-loops—promoter-paused R-loops and elongation-associated R-loops—that each display unique characteristics was proposed[57,58]. Promoter-paused R-loops are short R-loops frequently forming during promoter-proximal pausing of RNAPII[57,58]. R-loop mapping approaches based on RNaseH1 showed an enrichment of RNaseH1 at promoter-proximal sites[43,46]. It remains unclear whether and to which extent RNaseH1 binds to R-loops in other genomic regions. We therefore speculate that RDProx might be most sensitive in recovering proteins that associate with promoter-proximal R-loops. This could explain why some previously reported R-loop-associated proteins, such as the RNA/DNA helicase SETX and endonucleases XPG/XPF, were not identified in RDProx. For instance, SETX seems to primarily associate with R-loops at DSBs[59]. On the other hand, we would expect XPG and XPF in proximity to transcription-associated R-loops but it might be that these proteins are recruited only in occasions when R-loops are processed into DSBs and when not anymore bound by RNaseH1[19]. In addition, XPG and XPF are relatively low abundant in cells, which might preclude their identification by mass spectrometry. Similar might be true for the RNaseH2 complex, where we only identified RNaseH2A but not the B and C subunits of the complex.

Recent studies have shown that the RNA moiety in the hybrid can be modified with N6-methyladenosine (m6A)[14–16]. We now provide evidence that components of the m6A RNA machinery including the m6A writers (m6A–METTL-associated complex: VIRMA, ZC3H13, and RBM15), readers (hnRNPA2B1 and hnRNPC), and erasers (ALKBH5 and FTO) are indeed proximal to R-loops. This finding suggests that dynamic m6A deposition at RNA–DNA hybrids modulates the stability of R-loops in a context-dependent manner and through an interplay with other pre-mRNA processing factors. METTL3-dependent m6A deposition on RNA–DNA hybrids was shown to favor R-loop turnover during mitosis by recruiting the m6A reader YTHDF2[16]. A recent study identified a role for m6A RNA modification in stabilizing co-transcriptional R-loops forming at transcription termination sites, thereby ensuring faithful transcription termination and avoiding RNAPII read-through[15].

Another large group of proteins identified by RDProx was components of the DNA replication machinery such as the MCM complex, WDHD1, RFC1, MSH6, and CHAF1B. Regulation of R-loop levels by RNase H enzymes is known to be necessary for unperturbed replication[60,61]. Conversely, replication can influence R-loop formation during transcription–replication conflicts depending on the mutual orientation of the transcription and replication machineries: the replisome reduces R-loop levels when traveling co-directionally with the transcription machinery but stabilizes R-loops during head-on transcription–replication collisions[62]. The MCM complex was demonstrated to possess RNA–DNA helicase activity in vitro and is therefore potentially involved in the removal of R-loops in S phase[63]. Timely unloading of PCNA, as well as the recruitment of DEAD/DExH-box helicases to the replication fork, were shown to prevent replication-associated R-loop accumulation[64]. The identification of additional replication-associated factors by RDProx implies unexplored details of the crosstalk between DNA replication and R-loops.

Identification of SMARCA4, ARID1A, SMARCC1, and SMARCE1 proximal to R-loops and their known association with active transcription sites marked by H3K27 acetylation suggests a role of the SWI/SNF chromatin-remodeling complex in balancing R-loop levels[65,66]. The yeast and human FACT complex have been reported to resolve R-loop-mediated transcription–replication conflicts by reshaping the chromatin environment[67]. It has been recently reported that the SWI/SNF complex functions in resolving R-loop-mediated transcription–replication conflicts[68]. Furthermore, ARID1A-containing BAF complexes can recruit TOP2A to R-loop-associated chromatin, thereby preventing excessive R-loop formation and replication stress[69].

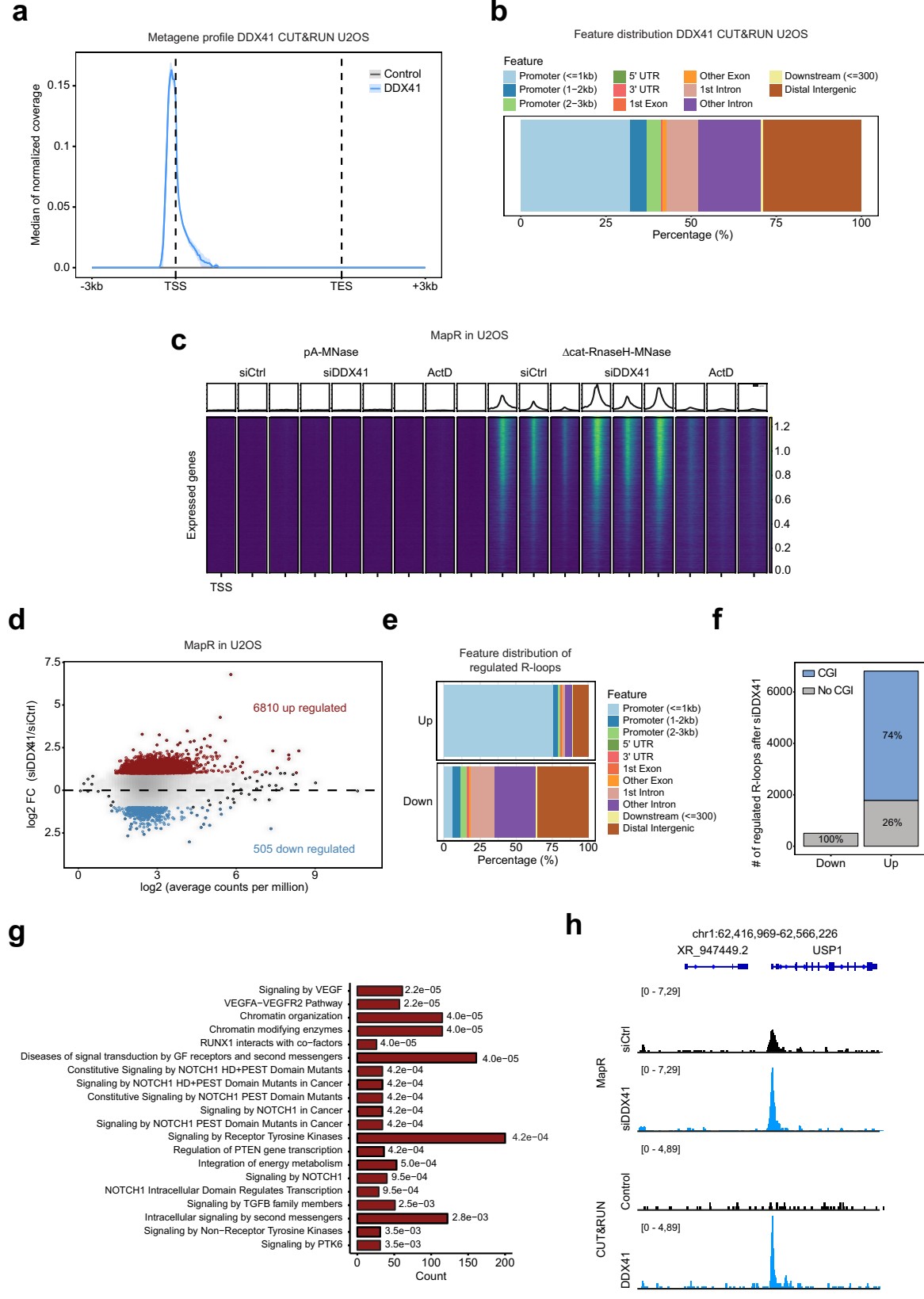

RDProx identified the role of the evolutionary conserved Dead box helicase 41 (DDX41) in opposing transcription-associated R-loop accumulation and DSBs at promoters. We demonstrate that DDX41 localizes to chromatin, preferentially associates with promoters, and opposes R-loop-dependent replication stress, DSBs, and genomic instability. Loss of DDX41 leads to a dramatic accumulation of R-loops in promoter regions—30% of all R-loops mapped to promoter regions accumulate upon loss of DDX41 in U2OS cells and this was even more apparent in HCT116 cells. Therefore, we propose that DDX41 plays a prominent role in counteracting the accumulation of R-loops in promoter regions of active genes. Dysregulated R-loops that obstruct the progression

**Fig. 4 Genome-wide analysis of DDX41 binding to chromatin and R-loops. a** Metagene profile showing the distribution of the GFP-DDX41 CUT&RUN signal in U2OS cells along expressed genes. **b** Genomic features overlapping GFP-DDX41 CUT&RUN peaks in U2OS cells. Features are color-coded as indicated in the legend. **c** MapR performed in $n = 3$ biologically independent experiments in U2OS cells after 48 h knockdown with control siRNA, DDX41 siRNA, or treatment with 4 μM Actinomycin D for 6 h. Heatmaps of normalized read coverage ranging from ±2 kb around the transcription start site of expressed genes sorted by gene expression based on the RNA-sequencing analysis of U2OS cells. **d** Scatter plot of MapR regions in U2OS cells. Consensus regions were constructed using the intersection of peaks for the replicates in each condition (siCtrl and siDDX41). The union of these regions was used for further analysis and quantification of the coverage/FC. The mean $\log_2$ fold change between siCtrl and siDDX41 is plotted against the $\log_2$ average counts per million representing the coverage. Genomic regions that are differentially regulated (FC > 2) are highlighted in red (up) or in blue (down). **e** Genomic feature distribution of the regulated MapR regions in U2OS cells after DDX41 knockdown. Features are color-coded as indicated in the legend. **f** The proportion of genomic regions with R-loop gains or losses in U2OS cells overlapping CGIs or not-overlapping regions are depicted. **g** Reactome pathway over-representation analysis for genes with R-loop gains in U2OS cells. The adjusted $p$-values (Fisher's exact test with Bonferroni-Holm correction) are indicated. **h** Representative snapshot of a genomic region depicting R-loops and GFP-DDX41 binding profiled by MapR and greenCUT&RUN, respectively, in U2OS cells.

of replication forks have been proposed as a major source of R-loop-dependent genomic instability[70]. Indeed, we found that DDX41 loss leads to genomic instability and increased fragility of DNA in promoter regions. We observed that DDX41 knockdown cells display slower replication fork progression and signatures of ATR-dependent signaling, suggesting that R-loop accumulation in DDX41 knockdown cells can obstruct the progression of replication forks and lead to the replication stress response. We also found that DDX41 knockdown cells show signs of perturbed transcription initiation and elongation. It is plausible that R-loop accumulation upon DDX41 loss leads to DSBs by interfering with replication and/or transcription machinery, which might explain why not all sites with R-loop accumulation display increased DNA fragility. It remains to be investigated by which mechanisms and under which conditions dysregulated R-loops are processed into DSBs. We also found sites of increased DNA fragility upon DDX41 loss that did not display R-loop accumulation and hence do not exclude a possibility that DDX41 safeguards actively transcribed genes through additional mechanisms.

RDProx also identified the DEAD-box helicase DDX39B/UAP56 that was described to participate in nuclear mRNA export as part of the TREX complex[27,71,72]. A recent study revealed the role of DDX39B in resolving R-loops by demonstrating that DDX39B/UAP56 associates with active transcription complexes to resolve R-loops throughout the gene body until the transcription termination site, thereby ensuring faithful transcriptional elongation and transcript release[34]. In contrast, DDX41 shows a striking preference to associate with, and oppose, R-loop accumulation in promoter regions, suggesting that different RNA–DNA helicases are required to balance R-loop levels in different genomic regions or after replication or transcription stress.

Interestingly, pathogenic variants in *DDX41* cause familial MDS/AML[39,41,73–75]. A recent study performed in zebrafish suggested that R-loop accumulation caused by Ddx41 deficiency leads to upregulated inflammatory signaling and aberrant expansion of the HSPCs[35]. Also, accumulation of R-loops was recently proposed as a feature of myelodysplastic syndrome (MDS) harboring splicing mutations[76–78]. In this work, we demonstrate that pathogenic variants in *DDX41* lead to the accumulation of DSBs in human HSPCs. Furthermore, we show that knockdown of DDX41 leads to the dependency of AML cells on ATR signaling (Fig. 6c). Genes that show R-loop accumulation are enriched for pathways frequently altered in AML such as chromatin organization, RUNX1 interactions as well as NOTCH and TGFβ signaling suggesting that DDX41 loss results in dysregulated transcription and aberrant cellular signaling through those pathways. These results suggest that pathogenic *DDX41* variants in human familial MDS/AML contribute to disease development through the accumulation of R-loops and DSBs as

well as provide incentives to explore ATR inhibition as a therapeutic strategy in these patients.

## Methods

**Cell culture.** U2OS, HCT116, and HEK293T cells were obtained from ATCC and cultured in D-MEM medium (U2OS and HEK293T) or RPMI 1640 (HCT116) supplemented with 10% fetal bovine serum, L-glutamine, penicillin, and streptomycin. OCI-AML3 cells were purchased from DSMZ GmbH and cultured in a D-MEM medium (PAN-Biotech) containing 20% FBS, L-glutamine, penicillin, and streptomycin. Cells were routinely tested for mycoplasma infection with a PCR-based method. For SILAC labeling, cells were cultured in media containing either L-arginine and L-lysine, L-arginine [13C6], and L-lysine [2H4] or L-arginine [13C615N4] and L-lysine [13C6-15N2] (Cambridge Isotope Laboratories)[79]. All cells were cultured at 37 °C in a humidified incubator containing 5% CO2.

**RDProx.** SILAC-labeled cells were transfected with a construct expressing APEX2-tagged HBD or HBD-WKK. After 48 h, cells were pre-treated with 500 μM biotin–phenol (Iris Biochem) for 2 h at 37 °C, followed by a 2 min incubation with 1 mM $H_2O_2$ (Sigma-Aldrich) at room temperature. Cells were washed twice with quenching solution (10 mM sodium azide, 10 mM sodium ascorbate, 5 mM Trolox (all from Sigma-Aldrich), and twice with PBS. Cells were lysed on ice using RIPA buffer (50 mM Tris, 150 mM NaCl, 0.1% SDS, 0.5% sodium deoxycholate, 1% Triton X-100). To release chromatin-bound proteins, cell lysates were sonicated using Bioruptor (Diagenode). For affinity purification of biotinylated proteins, equal amounts of differentially SILAC-labeled cell extracts, originating from either the HBD or the HBD-WKK condition, were combined prior to the pulldown and incubated with pre-equilibrated NeutrAvidin agarose beads (Thermo Scientific) for 2 h at 4 °C on a rotation wheel. Beads were washed once with RIPA buffer, thrice with 8 M Urea (Sigma) in 1% SDS, and once with 1% SDS in PBS. Bound proteins were eluted in NuPAGE LDS Sample Buffer (Life Technologies) supplemented with 1 mM DTT and boiled at 95 °C for 15 min. The eluates, after cooling down to room temperature, were alkylated by incubating with 5.5 mM chloroacetamide for 30 min in the dark and then loaded onto 4–12% gradient SDS-PAGE gels. Proteins were stained using the Colloidal Blue Staining Kit (Life Technologies) and digested in-gel using trypsin. Peptides were extracted from the gel and desalted on reversed-phase C18 StageTips.

**MS analysis.** Peptide fractions were analyzed on a quadrupole Orbitrap mass spectrometer (Q Exactive or Q Exactive Plus, Thermo Scientific) equipped with a UHPLC system (EASY-nLC 1000, Thermo Scientific) as described[80,81]. Peptide samples were loaded onto C18 reversed-phase columns (15 cm length, 75 μm inner diameter, 1.9 μm bead size) and eluted with a linear gradient from 8 to 40% acetonitrile containing 0.1% formic acid in 2 h. The mass spectrometer was operated in data-dependent mode, automatically switching between MS and $MS^2$ acquisition. Survey full-scan MS spectra ($m/z$ 300–1700) were acquired in the Orbitrap. The 10 most intense ions were sequentially isolated and fragmented by higher-energy C-trap dissociation (HCD)[82]. An ion selection threshold of 5000 was used. Peptides with unassigned charge states, as well as with charge states less than +2 were excluded from fragmentation. Fragment spectra were acquired in the Orbitrap mass analyzer.

**Peptide identification.** Raw data files were analyzed using MaxQuant (development version 1.5.2.8)[83]. Parent ion and $MS^2$ spectra were searched against a database containing 98,566 human protein sequences obtained from the UniProtKB released in 04/2018 using Andromeda search engine[84]. Spectra were searched with a mass tolerance of 6 ppm in MS mode, 20 ppm in HCD MS2 mode, strict trypsin specificity, and allowing up to 3 miscleavages. Cysteine carbamido-methylation was searched as a fixed modification, whereas protein N-terminal

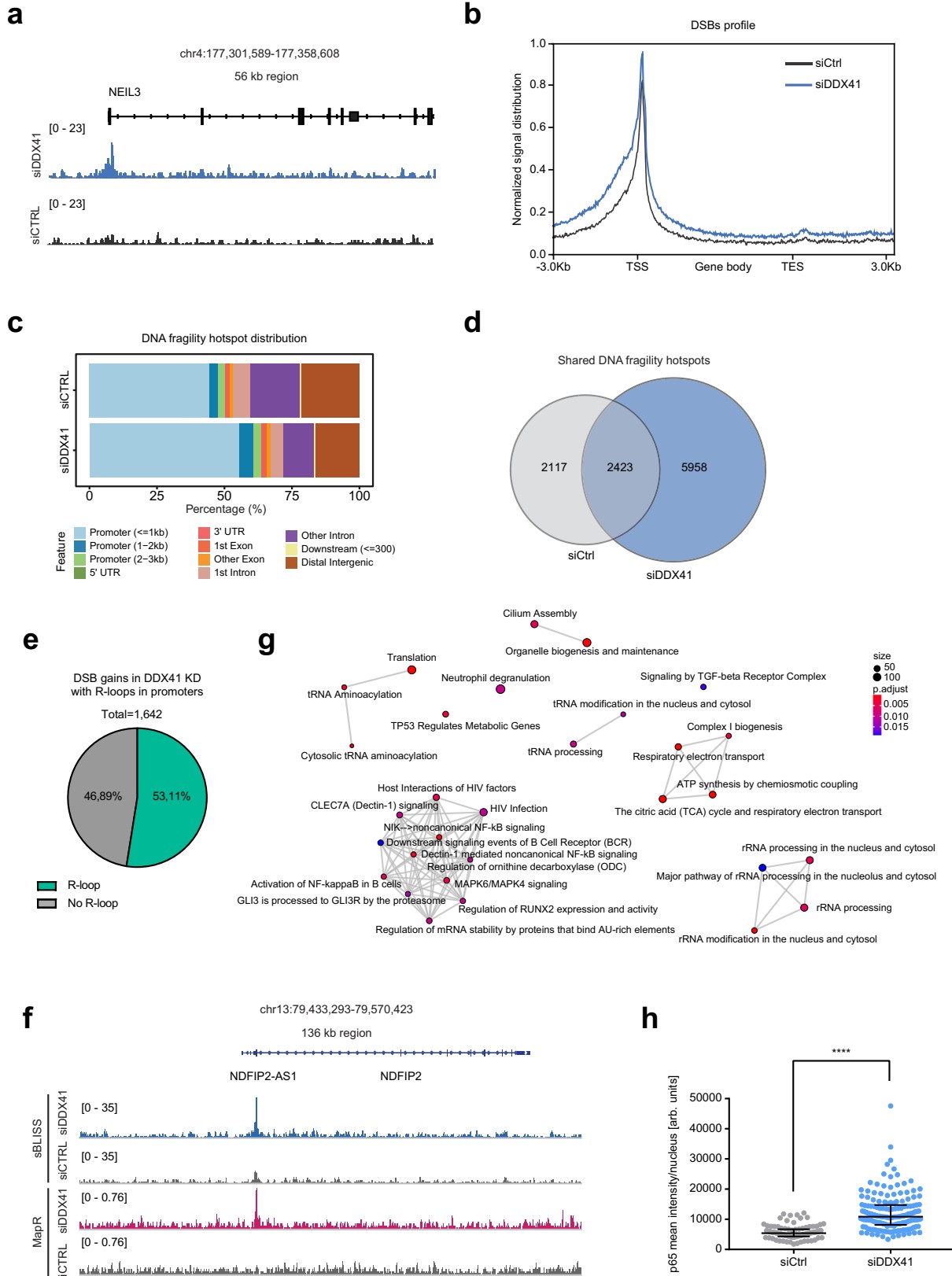

acetylation and methionine oxidation were searched as variable modifications. The data set was filtered based on posterior error probability (PEP) to arrive at a false discovery rate of below 1% estimated using a target-decoy approach[85].

**RDProx network analysis**. Pearson correlations were calculated using RStudio (version 1.3.959). Functional protein interaction network analysis was performed using interaction data from the STRING database[86]. Only interactions with a score >0.7 are represented in the networks. Cytoscape (version 3.2.1) was used for the visualization of protein interaction networks[87]. Genes were manually annotated by literature research and clustered based on similarity. PFAM domain enrichment analysis was performed using EnrichR[88]. The respective terms with the lowest FDR based on Fisher's exact test and correction for multiple comparisons are highlighted next to each cluster.

**Fig. 5 DDX41 loss leads to DSBs in promoters and inflammatory response. a** Representative snapshot of a genomic region depicting DNA fragility profiled by sBLISS in wild type and DDX41 knockdown HCT116 cells. **b** Metagene profile showing the double-strand break (DSB) signal distribution profiled by sBLISS along genes in wild type and DDX41 knockdown HCT116 cells. **c** Genomic features overlapping DNA fragility hotspots mapped by sBLISS in wild type and DDX41 knockdown HCT116 cells. Features are color-coded as indicated in the legend. **d** Venn diagram showing the number of unique and overlapped peaks mapped by sBLISS in wild type and DDX41 knockdown HCT116 cells. **e** Pie chart showing the percentage of double-strand breaks (DSB) gains (fold change (FC) > 2) mapped to promoters in DDX41 knockdown HCT116 cells that overlap or not with R-loops mapped in DDX41 knockdown (KD) HCT116 cells. **f** Representative snapshot of a genomic region showing accumulation of R-loops and DSBs profiled by MapR and sBLISS, respectively, in HCT116 cells. **g** Network of the Reactome pathway enrichment analysis of upregulated genes after DDX41 knockdown compared to control knockdown based on RNA-seq. All expressed genes were used as background. The size of the dots indicates the number of genes contributing to the displayed term. Gradual coloring represents the adjusted p-values based on Fisher's exact test with Bonferroni correction for multiple comparisons. **h** Immunofluorescence analysis of p65 after 48 h of indicated knockdowns in U2OS cells. Dots represent measurements of individual cells, black line indicates the median of the population with interquartile range. Representative data of $n = 2$ biologically independent experiments. $p$-value < 0.0001 derived from $n > 100$ cells using unpaired, two-sided Students's $t$-test. Source data are provided as a Source Data file.

**SDS-PAGE and western blotting**. Proteins were resolved on 4–12% gradient SDS-PAGE gels (NuPAGE® Bis-Tris Precast Gels, Life Technologies) and transferred onto nitrocellulose membranes. Membranes were blocked using 10% skimmed milk solution in PBS supplemented with 0.1% Tween-20. The list of antibodies used in this study and conditions can be found in Supplementary information. Secondary antibodies coupled to horseradish peroxidase (Jackson ImmunoResearch Laboratories) were used for immunodetection. The detection was performed with SuperSignal West Pico Chemiluminescent Substrate (Thermo Scientific).

**Neutral comet assay**. Neutral comet assay was performed according to the manufacturer's protocol (Trevigen). Briefly, cells were embedded in low melting agarose at 37 °C on Comet Slides (Trevigen). Overnight cell lysis at 4 °C was followed by equilibration in 1× Neutral Electrophoresis Buffer for 30 min at room temperature. Single-cell electrophoresis was performed at 4 °C in 1× Neutral Electrophoresis buffer for 45 min with constant 21 V. After DNA precipitation with 1× DNA Precipitation Buffer, Comet Slides were dried with 70% EtOH at room temperature. In order to completely dry the samples, Comet Slides were transferred to 37 °C for 15 min. DNA was stained with SYBR Gold solution for 30 min at room temperature. Images were taken with a Leica AF7000 microscope using a ×20 0.8NA air objective and a filter cube 480/40 nm, 505 nm, and 527/30 for excitation, dichroic, and emission wavelengths respectively. Tail moments of the comets were quantified using the CometScore (TriTek Corp.) software. At least 50 comets were quantified per condition.

**Quantification of RNA–DNA hybrids using dot blot**. Genomic DNA was extracted using the DNeasy mini kit (Qiagen). The isolated gDNA was treated with 1.2 U RNase III (produced in-house) for 2 h at 37 °C. After enzyme deactivation at 65 °C for 20 min, samples were split in half to digest control samples with 10 U RNaseH1 (NEB) overnight at 37 °C. Enzyme deactivation was followed by spotting DNA in a serial dilution on a nitrocellulose membrane (NeoLab Migge GmbH) using a dot-blot apparatus (BioRad). DNA was cross-linked to the membrane by UV light and afterward blocked with 10% skimmed milk solution in PBS supplemented with 0.1% Tween-20. The membrane was incubated overnight at 4 °C with the S9.6 antibody (produced in-house). After incubation of secondary antibodies conjugated to horseradish peroxidase (Jackson ImmunoResearch Laboratories) signal was detected using SuperSignal West Pico Chemiluminescent Substrate (Thermo Scientific). An antibody against dsDNA was probed as a loading control after stripping the membrane with β-mercaptoethanol (Sigma) and 0.1% SDS in PBS. The detected signal was quantified using Fiji/ImageJ (v1.51) and ratios between the signal resulting from S9.6 and dsDNA staining were calculated to quantify global R-loop levels[89].

**Proximity ligation assay**. Proximity Ligation Assay was performed according to the manufacturer's protocol (Duolink®, Sigma-Aldrich). Cells were fixed with 4% paraformaldehyde in PBS and permeabilized with 0.25% Triton X-100. Samples were blocked with Duolink® Blocking Solution for 1 h at 37 °C in a humidity chamber. After removal of the blocking solution, primary antibodies diluted in Duolink® Antibody Diluent were added on the coverslips for 2 h at room temperature in a humidity chamber. Coverslips were washed 2× with Washing Buffer A. PLA plus and minus probes were put on in a 1:5 dilution in Duolink® Antibody Diluent for 1 h at 37 °C in a humidity chamber. Two washes with Washing Buffer A were followed by Ligase treatment in 1× Ligation Buffer for 30 min at 37 °C in a humidity chamber. Ligation buffer was tapped off and coverslips were washed twice with Washing Buffer A. Amplification was achieved by adding the Polymerase in 1× Amplification buffer for 100 min at 37 °C in a humidity chamber. After washing the samples 2× with 1× Washing Buffer B and 1× with 0.01× Washing Buffer B, coverslips were stained with 1 µg/ml Hoechst33342 and mounted using Dako mounting medium. Images were taken with a Leica SPE

microscope using a ×63 1.4NA oil objective. The number of PLA spots per nucleus was quantified using Fiji/ImageJ (v1.51)[89].

**ATPase assay**. The ADP-Glo Assay was performed according to the manufacturer's protocol (Promega). In brief, an ATP/ADP standard curve was prepared before each experiment in order to interpolate the measured values. Purified full-length DDX41 was incubated in a serial dilution together with 100 nM of RNA–DNA substrate with an ssDNA overhang and 5 µM ATP. After incubating the mix at 37 °C for 60 min, the reaction was stopped by depleting unconsumed ATP with the ADP-Glo Reagent. The Kinase Detection Buffer was added to convert ADP to ATP and to add luciferase and luciferin to detect ATP. The resulting luminescence was measured with a Spark M200 (Tecan). The measured values were interpolated based on the values obtained by the ATP/ADP standard curve using GraphPad PRISM (v7.04, Graphpad Software, Inc.).

**Fluorescence polarization assay**. DsDNA, dsRNA, and RNA–DNA hybrid 12-mer substrates were generated by heating the respective 6-FAM-conjugated and unlabeled oligonucleotide pairs to 95 °C and gradually cooling them down to 4 °C. Single-stranded and double-stranded substrates were diluted to a final concentration of 20 nM in FP assay buffer (20 mM HEPES, pH 7.0, 100 mM NaCl, 5% glycerol). Purified full-length DDX41 protein, HBD, or HBD-WKK were added to the individual substrates in a serial dilution. Fluorescence polarization of the 6-FAM-labeled probes was analyzed on a Tecan Spark 20 M plate reader at 20 °C (excitation wavelength: 495 nm, emission wavelength: 520 nm, gain: 100, flashes: 15, integration time: 40 µs). Relative fluorescence polarization was calculated by subtracting the FP value of the oligo-only conditions. Binding constants (Kd values) were determined by fitting a Michaelis–Menten non-linear regression onto the relative FP values in GraphPad Prism (v7.04, Graphpad Software, Inc.).

**Electrophoretic mobility shift assay**. 20 nM of 6-FAM-conjugated single- and double-stranded oligonucleotides were incubated with 25 µM of purified HBD or HBD-WKK mutant for 10 min at room temperature in interaction buffer (20 mM Tris-Cl pH 7.5, 150 mM NaCl, 10% glycerol, 1 mM EDTA, 1 mM DTT). 6× loading buffer (60% Glycerol, 20 mM Tris-Cl pH 8.0, 60 mM EDTA) was added to the samples before loading them on a 20% Novex TBE gel (ThermoFisher Scientific). The gel was run for 45 min at 200 V in TBE buffer and scanned using a Typhoon FLA 9000 @ 473 nm to visualize the fluorescence of the 6-FAM-labeled probes.

**FRET-based unwinding assay**. RNA–DNA hybrid substrates with a single-stranded DNA overhang were generated by mixing an IBFQ-conjugated 38-mer DNA oligo (IDT) and a 6-FAM-conjugated 13-mer RNA oligo (IDT) and heating them to 95 °C and gradually cooling them down to 4 °C. Annealed substrates were incubated together with 5 µM ATP and either full-length DDX41 or mutant proteins. Increased fluorescence intensity upon addition of DDX41 after displacement of the quencher during unwinding was measured on a Spark M20 (Tecan) plate reader.

**qPCR analysis**. RNA was extracted using the RNeasy Plus Mini Kit (Qiagen). 500 ng of purified RNA was reverse-transcribed into cDNA by using the QuantiTect Reverse Transcription Kit (Qiagen). Purified cDNA was amplified during qPCR on a CFX384 BioRad instrument using 2× SYBR Green mix and 0.5 µM final primer mix.

**Cell viability assay**. Cell viability assay was performed using the Cell Titer-Blue Cell Viability Assay (Promega) according to the manufacturer's instructions.

**RNA-sequencing and data analysis**. RNA was extracted using the RNeasy Plus Mini Kit (Qiagen). In brief, cells were lysed and genomic DNA was depleted. Samples were treated with DNase to remove residual DNA. After purification using

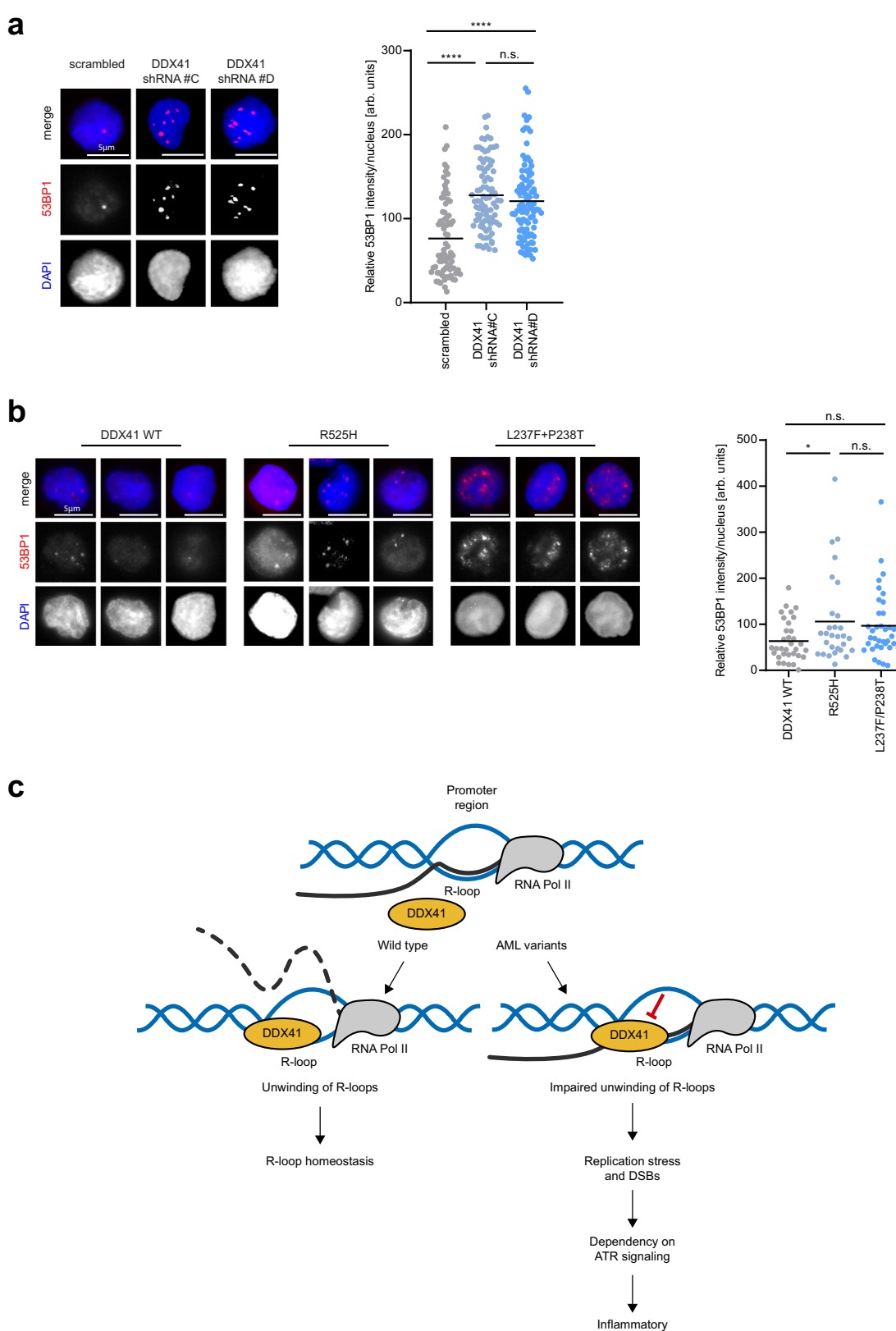

spin columns, RNA was eluted in RNase-free water and stored at −80 °C until library preparation. NGS library prep was performed with Illumina's TruSeq stranded mRNA LT Sample Prep Kit following TruSeq Stranded mRNA Reference Guide (Oct.2017) (Document # 1000000040498v00). Libraries were prepared with a starting amount of 1000 ng and amplified in 10 PCR cycles. Libraries were profiled in a High Sensitivity DNA on a 2100 Bioanalyzer (Agilent Technologies) and quantified using the Qubit dsDNA HS Assay Kit, in a Qubit 2.0 Fluorometer

(Life Technologies). All 15 samples were pooled in equimolar ratio and sequenced on a NextSeq 500 Highoutput FC, SR for 1 × 84 cycles plus 7 cycles for the index read. All genomic libraries were sequenced on an Illumina NextSeq 500 and de-multiplexed using blc2fastq (v2.19). RNA-seq samples were sequenced with a read length of 84 bp in single read mode. Samples were mapped using STAR (v2.7) against hg38 with the Gencode annotation (v25)[90,91]. Reads per gene were counted using featureCounts (v.1.6)[92]. The differential expression analysis was performed

**Fig. 6 Model for DDX41 function in R-loop homeostasis. a** Immunofluorescence analysis of HSPCs after 24 h of indicated knockdowns. Cells were nucleofected with plasmids encoding GFP and respective shRNAs. GFP-positive cells were sorted via FACS and seeded on coverslips. Representative images of 53BP1 (red) staining in HSPCs (left). DNA was counterstained with DAPI (blue). Quantification of nuclear 53BP1 intensity (right). Each dot represents a single measured value. The black line indicates the median. At least 80 cells across $n = 2$ biologically independent experiments were measured per condition. $p$-values ($p < 0.0001$, $p < 0.0001$, $p = 0.2227$) were calculated by one-way ANOVA with Tukey correction for multiple comparisons. Scale bars—20 µm. Source data are provided as a Source Data file. **b** Immunofluorescence analysis of HSPCs after expression of DDX41 WT, L237F + P238T or R525H mutants tagged with GFP. GFP-positive cells were sorted by FACS and used for the analysis. Representative images of 53BP1 (red) staining in HSPCs (left). DNA was counterstained with DAPI (blue). Quantification of nuclear 53BP1 intensity (right). Dots represent results from individual cells. The median is indicated by the black line. $p$-values ($p = 0.0237$, $p = 0.624$, $p = 6.018$) were derived from at least 30 cells across $n = 1$ experiment using one-way ANOVA with Tukey correction for multiple comparisons. Scale bars—20 µm. **c** Wild-type DDX41 associates with R-loops in promoters of active genes and balances R-loop levels by unwinding RNA–DNA hybrids. Pathogenic DDX41 variants found in acute myeloid leukemia (AML) display impaired RNA–DNA hybrid unwinding activity, leading to the accumulation of R-loops at promoters. Accumulation of R-loops at promoters results in increased replication stress, DSBs, and inflammatory signaling, rendering DDX41 mutated AML cells vulnerable to ATR inhibitors.

---

using Bioconductor (v2.46)/DESeq2 (v1.26)[93,94]. Genes were deemed significantly differentially regulated with an FDR below 1%. Coverage tracks were normalized and created using deepTools (v3.4.1)[95]. Genes were deemed expressed within the analysis if they were tested for differential expression in the DESeq2 analysis. We used GSM2296622 to generate a list of expressed genes for the HCT116 cell line. The raw data were downloaded from GEO and mapped using STAR against hg38 with Gencode annotation. Reads per gene were counted using featureCounts.

**MapR**. MapR was performed according to the before published protocol with minor modifications[46,47]. Cells were either treated with indicated siRNAs or with 4 µM Actinomycin D (Cell Signaling Technology) for 6 h. Concanavalin-A-coated beads (Polysciences Europe GmbH) were activated in Binding Buffer (20 mM Hepes–KOH pH 7.9, 10 mM KCl, 1 mM CaCl$_2$, 1 mM MnCl$_2$). 5*10$^5$ U2OS cells were washed twice with Wash Buffer (Hepes-NaOH pH 7.5, 150 mM NaCl, 0.5 mM Spermidine, 1 mM protease inhibitor) at room temperature and afterward immobilized on the activated beads in 50 µl Wash Buffer containing 0.05% Digitonin. Either pA-MNase or RHΔ-MNase was added to the cells overnight at 4 °C on a rotating wheel. After three washes with Wash Buffer containing 0.05% Digitonin (Millipore), samples resuspended in 100 µl Dig-Wash-Buffer were equilibrated on ice. The activity of the MNase was triggered by adding 2 mM CaCl$_2$ to the samples for 30 min. 2× Stop Buffer (68 µl 5 M NaCl, 40 µl 0.5 M EDTA, 20 µl 0.2 M EGTA, 10 µl 5% digitonin, 5 µl 10 mg/ml RNase A) was mixed with the samples to stop the reaction. Chromatin fragments were released by incubating the samples for 20 min at 37 °C and centrifugation at 16.000×g for 10 min at 4 °C. Supernatants were incubated at 70 °C in the presence of 0.1% SDS and 5 µg proteinase K. Before library preparation, the DNA was recovered by phenol-chloroform extraction. NGS library preparation was performed using NuGEN's Ovation Ultralow System V2 (M01379 v5). Libraries were prepared with a starting amount of 1 ng of DNA and were amplified in 12 PCR cycles. Libraries were profiled in a High Sensitivity DNA on a 2100 Bioanalyzer (Agilent Technologies) and quantified using the Qubit dsDNA HS Assay Kit, in a Qubit 2.0 Fluorometer (Life technologies). All 18 samples were pooled in equimolar ratio and sequenced on one NextSeq 500 High output Flowcell, PE for 2 × 42 cycles plus 8 cycles for the index read.

**MapR analysis**. U2OS as well as HCT116 MapR samples were mapped against h38 using bowtie2 (v2.3.4) the result was filtered for uniquely mapping reads[96]. Peak calling was performed using MACS2 (v2.1.2)[97] with the parameters "--keep-dup auto --broad --broad-cutoff 0.1 --bw 100 --min-length 100 --format BAMPE --g hs". The MapR samples were further analyzed using the FC between siDDX41 and siCtrl. A set of consensus regions was created using the intersection of peaks called per group (either in siDDX41 or siCtrl replicates). Then the union of these two peak sets was used to quantify the signal present in the samples. Using R/Bioconductor[94] packages the fold change for the consensus regions was calculated using the average normalized coverage (rpkm) of the regions. Normalization was based on the total amount of sequenced reads. R-loop gains were determined based on the FC > 2 in siDDX41 compared to siCtr U2OS cells and FC > 1.5 in siDDX41 compared to siCtr HCT116 cells. Sequencing depth normalized coverage tracks for all samples and metagene/enrichment around the TSS or from TSS to TES were created using deepTools (v3.4.1)[95] and further processed using custom R scripts.

**sBLISS and data analysis**. sBLISS in HCT116 cells was performed as previously described[48,49] with the following modifications: After blunting of DNA DSB ends in fixed nuclei, samples were 3′ adenylated using Klenow Fragment (3′-->5′ exo-) (NEB M0212) at a final amount of 60U per reaction in 1× NEBNext® dA-Tailing Reaction Buffer (NEB B6059). The A-tailing reaction was incubated at 37 °C for 1 h in a thermo-shaker at 300 rpm. Fixed nuclei were then washed 2× with CST buffer (CutSmart buffer B7204 supplemented with 0.1% Triton) in order to washout the Klenow enzyme. DNA DSB-end labeling was performed as described[48] with the following modification: sBLISS linkers containing one thymine overhang at the 3′ end of the reverse oligo were used. Downstream sample processing steps were

carried out with 150 ng of DNA template input from each sample for in vitro transcription reaction. sBLISS data were processed as described previously[48] using GRCh38/hg38 reference genome with BWA-MEM[98] (version 0.7.15). We used MACS2[97] (version 2.2.6) to call peaks from the BED files of UMI-DSB as reported previously[49]. Peaks identified by MACS2 with $q$-value < 0.01 were annotated using Chipseeker[99] (version 1.22.0). Peaks lists from both conditions were merged using bedtools[100] (version 2.27.0). The count-per-million (CPM) values for the merged peaks were calculated and normalized by library size with edgeR[101] (version 3.32.1). The peaks with gain and loss of breakage were classified based on the fold change greater than 2 between siCTRL and siDDX41 samples.

**greenCUT&RUN and data analysis**. CUT&RUN was performed in a stable U2OS cell line that expresses N-terminally GFP-tagged DDX41 under a doxycycline-inducible promoter. Expression of GFP-DDX41 was induced by adding 1 µg/ml doxycycline for 48 h or DMSO for un-induced control cells. Cells were mildly crosslinked with 1% formaldehyde for 2 min at room temperature. Quenching of the reaction with 125 mM glycine was followed by cell detachment using trypsin and two subsequent washes in Wash buffer (20 mM HEPES–KOH (pH 7.5), 150 mM NaCl and 0.5 mM spermidine and EDTA-free complete protease inhibitor). Concanavalin-A beads were activated in binding buffer (20 mM HEPES–KOH (pH 7.9), 10 mM KCl, 1 mM CaCl$_2$, and 1 mM MnCl2) for 5 min at room temperature and afterward 1*10$^6$ cells immobilized on the beads for 10 min at room temperature. After cell permeabilization with 0.05% digitonin-Wash buffer, 1 µg of GFP-nanobody-MNase (GFP-nanobody LaG16 described in ref. [44]) was added in 100 µl and incubated with the immobilized cells at 4 °C for 30 min. Unbound MNase was washed out two times with digitonin-Wash buffer before transferring samples to an ice bath. The MNase was activated by addition of 3 mM CaCl2 for 30 min and the reaction was subsequently stopped by adding 2× Stop buffer (340 mM NaCl, 20 mM EDTA, 10 mM EGTA, 0.05% digitonin, 100 µg/ml of RNase A, and 50 mg/ml glycogen). DNA fragments were released for 20 min at 37 °C before de-crosslinking overnight at 55 °C in the presence of 0.1% SDS and 1.5 µl of 20 mg/ml proteinase K. DNA was cleaned-up by phenol-chloroform extraction before subsequent library preparation using the Accel-NGS 1S Plus DNA Library Kit (Swift Bioscience) according to the manu-factures' protocol suggested for the retention of small fragments (>40 bp). Libraries were dual indexed and amplified for 14 cycles (NEBNext Multiplex Oligos for Illumina, Dual Index Primers Set 1). An equimolar pool of libraries was prepared and further purified away from primer and adaptor dimers on a 2% agarose gel using the Zymoclean Gel DNA Recovery Kit (Zymo). Final quantification and quality control before sequencing was done on an Agilent 2200 TapeStation System. Samples were sequenced within paired read mode with 34 bp in read 1 and 49 bp in read 2. The first 15 bases of the second read of the cut and run data were removed and the data was adapter trimmed using Cutadapt (v.1.18). The data was mapped against hg38 using bowtie2 (v.2.3.4)[96] and filtered for uniquely mapping reads. Peak calling was done using MACS2 (v2.1.2)[97] with the following parameters "-g hs --min-length 150 --format BAMPE --keep-dup auto".

**DNA fiber spreading assay**. U2OS cells were labeled with 5-Chloro-2′-deoxy-yuridine (CldU, 30 µM) for 30 min, washed once with warm PBS, then labeled for 30 min with 5-Iodo-2′-deoxyuridine (IdU, 340 µM). Cells were either transfected with siDDX41/siCTRL for 24 h or treated with 0.1 µM aphidicolin (APH) for 1.5 h. After labeling, cells were washed once with warm and 3× with cold PBS, then trypsinized and spun down (300×g, 5 min). They were resuspended in cold PBS, counted and diluted to 5 × 10$^5$/ml. Labeled cells were diluted with twice the number of unlabeled cells. 4 µL of the cell suspension were mixed with 7.5 µl of the lysis buffer (200 mM Tris-HCl pH 7.4, 50 mM EDTA, 0.5% SDS) directly on the SuperFrost Plus microscopy slide (Thermo Scientific) and incubated horizontally for 9 min. The slides were then tilted at 30°−45°, allowing DNA fibers to spread to the bottom of the slide. DNA spreads were air-dried and fixed with 3:1 metha-nol:acetic acid overnight at 4 °C. The fibers were rehydrated 3 × 3 min in PBS, dipped once in Milli-Q water and denatured in 2.5 M HCl for 1.5 h at RT, then

washed 5 × 3 min in PBS. The slides were blocked for 40 min in the blocking solution (2% BSA, 0.1% Tween-20 in PBS) and incubated with primary antibodies (mouse anti-BrdU, 1:100, BD Bioscience and rat anti-BrdU, 1:500, Abcam) at RT for 2.5 h. After 3 × 5 min washes with PBS-T, the slides were incubated with secondary antibodies (goat anti-mouse Cy3.5, Abcam and goat anti-rat Cy5, Abcam) at RT in the dark for 1 h. The spreads were washed 3 × 5 min with PBS-T, dipped in Milli-Q water, and air-dried completely in the dark. The slides were mounted using Prolong Gold AntiFade mountant (Thermo Scientific), imaged with Visiscope 5-Elements Spinning Disc Confocal microscope (Visitron Systems, Germany) (magnification: ×60 Water immersion objective with ×2 extra magnification; laser lines and corresponding emission filters: 640 nm, 692/40 and 561 nm 623/32) and quantified using the Fiji/ImageJ software[89].

**Immunofluorescence**. Cells were washed 2× with PBS, incubated with 0.4% NP-40 for 20 or 40 min on ice, and washed 2× with PBS-T (0.1%). Cells were fixed with 4% paraformaldehyde in PBS for 15 min at room temperature, washed 2× with PBS-T (0.1%), and permeabilized with Triton X-100 (0.3%) for 5 min at room temperature, followed by 2× washes with PBS. Cells were blocked for 1 h with 5% fetal bovine serum albumin in PBS-T (0.1%) containing penicillin and streptomycin. Incubation with primary antibodies diluted in blocking buffer was performed overnight at 4 °C and followed by 3× washes with PBS-T and 1 h incubation with Alexa Fluor-coupled secondary antibodies in a dark chamber at room temperature. Nuclei were counterstained with 1 µg/ml Hoechst33342 in PBS either simultaneously with secondary antibody incubation or for 30 min. For chromatin retention assay permeabilization, blocking and antibody incubations steps were omitted. Cells were washed 2× with PBS-T and kept at 4 °C in PBS until imaging. Imaging was performed with an Opera Phenix (PerkinElmer) microscope using a ×40 1.1NA water objective. Image analysis was performed by using Harmony High-Content Imaging and Analysis Software (version 4.4, PerkinElmer). Standard building blocks allowed for nuclei segmentation based on the Hoechst signal and cells on the edges of the field were excluded. Mean intensity measurements were performed for maximum projections and spot detection was calculated by using algorithm B.

**Hematopoietic stem and progenitor cell experiments**. For *DDX41* knockdown experiments primary human CD34+ cells (from cord blood, purchased from Lonza) were transfected with two different shRNA constructs (TL305064C; GCTATGCAGACCAAGCAGGTCAGCAACAT; TL305064D; GCGTGCGGAA GAAATACCACATCCTGGTG) (Origene) using Amaxa® Human CD34+ Cell Nucleofector® Kit according to the manufacturer's recommendations (Lonza). Similarly, for ectopic expression of *DDX41* WT, *DDX41* L237F P238T, and *DDX41* R525H, respectively, primary human CD34+ cells (from cord blood, purchased from Lonza) were transfected using Amaxa® Human CD34+ Cell Nucleofector® Kit according to the manufacturer's recommendations (Lonza). Cells were cultured in StemSpan SFEM II supplemented with myeloid expansion supplement containing SCF, TPO, G-CSF, and GM-CSF (Stemcell Technologies) for 24 h before isolation of GFP+ cells with a BD FACSMelody cell sorter, using double sorting to ensure maximum purity (BD Biosciences). For immunofluorescence experiments, cells were seeded onto glass slides, fixed with 4% PFA for 10 min, permeabilized using 0.15% Triton X-100 for 2 min, and blocked in 1% BSA/PBS. 53BP1 was detected using anti-53BP1 (nb100-904; Novus Biologicals) followed by Alexa Fluor® 594-conjugated goat anti-rabbit antibody (ThermoFisher Scientific). Slides were mounted in VectaShield containing 1 µg/ml DAPI (Vector Laboratories). Images were acquired on a DMi8 Leica inverted microscope (×100 objective) and processed using LasX software (Leica). Quantification of mean fluorescence intensity (MFI) was performed using the following formula: mean fluorescence of selected cell—(area of selected cell × mean fluorescence of background readings). Values are displayed as arbitrary units (A.U.). In order to assess *DDX41* knockdown efficiency, about 5000 eGFP+ cells were sorted in RLT buffer (Qiagen) for RNA extraction followed by cDNA synthesis. DDX41 expression was quantified using qRT-PCR (DDX41-fw-qPCR, DDX41-rev-qPCR).

**Protein purification**. His6-3C-DDX41 full-length WT and R525H were expressed in SF9 insect cells using the Bac-to-Bac system and SF900 III media (ThermoFisher). His6-DDX41 (153–410), His6-GST-3C-RNase H (27–76 = HBD)-AVI-tag WT, W43/K59/K60-A (WKK-A) (all pET28) and pelB-GFP-nanobody (LaG16)-MNase-His6-HA (pET21b, LaG16 nanobody described in X) were expressed in *E. coli* (BL21 DE3 codon+) using LB media. Cells were lysed in lysis buffer (30 mM Tris-Cl pH 8.0, 500 mM NaCl, 1 mM MgCl2, 1 mM DTT, 5% glycerol, 100 U/ml Benzonase, EDTA-free cOmplete protease inhibitor cocktail, 15 mM imidazole except for the HBD constructs, 0.5 % Triton X-100 for the DDX41 full-length constructs), using a Branson Sonifier 450 and cleared by centrifugation (40,000×g, 30 min at 4 °C). In case of the HBD constructs, additional 500 mM NaCl was added to the cleared lysates and a PEI-based precipitation of nucleic acids (0.2% w/v polyethylenimine, 40 kDa, pH 7.4) for 5 min at 4 °C was performed, followed by a second round of centrifugation (4000×g, 4 °C, 15 min). Recombinant proteins were affinity-purified from cleared lysates using a NGC Quest Plus FPLC system (Biorad) and Cytiva columns: HisTrap FF crude (DDX41 fl variants), HisTrap FF 5 ml (DDX41(153–410), LaG16-MNase), GSTrap HP 5 ml (HBD variants), following the manufacturer's protocols. DDX41(153–410) and LaG16-MNase were further subjected to Heparin-based

chromatography (HiTrap Heparin HP 5 ml, Cytiva, in 30 mM Na-Hepes, 25 mM NaCl, 5% glycerol) following the manufacturer's protocol. DDX41 fl variants and HBDs were digested with His6-3C protease (1:100 w/w) overnight at 4 °C in the presence of 1 mM DTT to cleave off the His6- and His6-GST tag, respectively. Digested HBDs were run over a HisTrap ff 5 ml column (Cytiva) to absorb out the His6-GST and His6-3C protease. All recombinant proteins were concentrated using Amicon spin concentrators (Merck Millipore) and subjected to gel filtration (in 30 mM Na-Hepes, 300 mM NaCl, 1 mM DTT, 10% Glycerol, pH 7.5, additional 1 mM EDTA for HBDs and Lag16-MNase). DDX41 fl variants were run twice on a Superdex 200 16/60 pg (Cytiva), all other proteins were run once on a Superdex 75 16/60 pg (Cytiva). Peak fractions containing the recombinant proteins after gel filtration were pooled and protein concentration was determined by using absorbance spectroscopy and the respective extinction coefficient at 280 nm, before aliquots were flash frozen in liquid nitrogen and stored at −80 °C.

**Flow cytometry**. For the HBD-GFP retention assay, HBD-GFP expression in U2OS cells was induced with 1 µg/ml doxycycline (Sigma-Aldrich) for 48 h. Cells were collected with trypsin and pre-extracted with 0.05% Triton X-100 in PBS for 3 min at room temperature. After fixation with 4% PFA and subsequent PBS washes, cells were analyzed on an LSRFortessa SORP (Becton Dickinson). 30,000 cells were measured. The 488 nm laser and 530/30 band pass filter were used for analyzing GFP fluorescence. DNA content was analyzed using 1 µg/ml Hoechst33342 (Excitation 355 nm, Emission 450/50 BP). Downstream data analysis was performed with FlowJo (v10.5.3, Becton Dickinson). After gating for singlets (FSC-A/FSC-H, then Hoechst-A/Hoechst-H), the proportion of GFP-positive cells was quantified by setting a threshold above the fluorescence of DMSO-treated control cells.

**Fluorescence-activated cell sorting**. Expression of N-terminally GFP-tagged DDX41 WT, L237F + P238T or R525H in OCI-AML3 cells was induced with 3 µg/ml doxycycline (Sigma-Aldrich). 72 h after induction, cells were spun down and washed twice with PBS. After re-suspending cells in PBS, they were sorted by FACS using a 100 µM nozzle on a BD FACSAria III SORP (Becton Dickinson) in purity precision mode with FACSDiva software version 8.0.2. Cells of interest were identified via FSC-A/SSC-A. Subsequently, doublets were excluded via FSC-A/FSC-H and dead cells were excluded by DAPI staining (0.5 µg/ml final concentration) using a 405 nm laser and 450/50 BP. GFP cutoff was set according to non-expressing cells. Sorting based on GFP was achieved using the 488 nm laser and 530/30 band pass filter. Roughly 500,000 cells were sorted directly into fresh a-MEM containing 20% FBS and further cultured until subsequent experiments.

**Reporting summary**. Further information on research design is available in the Nature Research Reporting Summary linked to this article.

## Data availability
The data that support this study are available from the corresponding author upon reasonable request. The fasta file containing the human reference proteome (released in 04/2018) used for the analysis of raw MS data using MaxQuant was retrieved from UniprotKB UP000005640. HCT116 GRO-seq data to determine gene expression levels for MapR analysis was retrieved from GEO with access code GSM2296622. RNA-Seq, MapR, and BLISS data generated in this study have been deposited in the GEO database under accession code GSE168173. The mass spectrometry-based proteomics data have been deposited to the ProteomeXchange Consortium via the PRIDE partner repository[102] with the data set identifier PXD024517. Source data are provided with this paper.

## Code availability
Any custom code is available upon request.

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

## Acknowledgements

We thank the members of the Beli group for helpful discussions, and Helle Ulrich and Nicola Zilio (IMB) for providing the GFP-nanobody fused to MNase. This work is funded by the Deutsche Forschungsgemeinschaft (DFG, German Research Foundation)—Project-ID 393547839—SFB 1361 and Project-ID BE 5342/2-1—FOR 2800 awarded to P.B. and Project-ID 393547839—SFB 1361, Project-ID 402733153—SPP 2202 and Project-ID 455784893 awarded to V.R. Support by the IMB Genomics Core Facility and the use of its NextSeq 500 funded by the Deutsche Forschungsgemeinschaft (DFG, German Research Foundation)—INST 247/870-1 FUGG) is gratefully acknowledged. We thank Jimmy Chen and Amitkumar Fulzele for assistance with mass spectrometry analysis, and Andrea Voigt and Katharina Mayr for technical support. Support by the IMB microscopy and flow cytometry core facility is gratefully acknowledged. VisiScope Spinning Disc Confocal microscope (Project number 402386039) and the Opera Phenix (INST 247/845-1 FUGG) are funded by the DFG.

## Author contributions

P.B., T.M., V.R., and B.L. designed the research. T.M. and F.C. performed experiments and analyzed the data. I.M. performed a DNA fiber assay and analysis. N.K. performed the analysis of MapR and RNA-sequencing. G.M.C.L. performed sBLISS. G.P. performed the analysis of sBLISS. M.M. purified all recombinant proteins and helped with fluorescence polarization and EMSA assays. J.F. conducted the experiments in CD34+ HSPCs. J.B. helped with establishing conditions and performing CUT&RUN. P.B. and T.M. wrote the manuscript. All authors read and commented on the manuscript.

## Competing interests

The authors declare no competing interests.
