## [Peer Review File · Nature Communications]

R-loop proximity proteomics identifies a role of DDX41 in transcription-associated genomic instabilityReviewers' Comments:

Reviewer #1:

Remarks to the Author:

This manuscript describes an R-loop proteomics screen followed after a hard pivot by the characterization of the genome instability defects caused in human cells by deficiency in the Ddx41 helicase. The manuscript has its strengths and is generally well-presented and easy to read. It suffers, however, from significant shortcomings that are detailed below. Maybe the biggest flaw as things stand now is that the data is not particularly novel and is limited to a correlative analysis of the relationship between R-loops and phenomena of genome instability.

One shortcoming of the manuscript is that it doesn't acknowledge, and therefore oversells, the proteomics screen. The issue here is that the use of the HBD domain as an R-loop sensor guarantees that only a subset of R-loops are effectively targeted. These correspond primarily to short R-loops that form at paused promoters mostly corresponding to CpG islands. This reflects a recent proposal (PMID: 33411340) that simply takes into account the vast differences between in situ mapping approaches (typically RNase H1-based like MapR) and ex vivo mapping approaches (typically S9.6-based like DRIP). I request that the authors more specifically discuss this issue and highlight that the screen is most sensitive to a subset of R-loops mapping primarily to paused promoters. This matters much for making precise claims and also serves to explain the results of the screen. Promoter-proximal pausing is a well-established quality control step that permits the coordination of transcription with vital RNA processing complexes such as the spliceosome and RNA export machinery. Not surprisingly, these complexes are the top scoring complexes. Similarly, CpG island promoters are well-known to associate with replication origins (for instance PMID: 19360092) which could account for the observed links with replication proteins. Finally, the pause site are located right before the highly dynamic +1 nucleosome which could explain the identification of chromatin remodeling complexes. Overall, this goes to say that the regio-specificity inherent to the screen should be directly acknowledged and discussed better than it is now.

The proteomics screen is interesting but given very short change in the manuscript. What shines brightly from the results is the absence of many factors hypothesized to play key roles in R-loop metabolism including many mentioned in the Introduction. For instance, SETX does not appear here. Nor do GADD45A and TET1 even though they were supposed to be recruited to R-loops at CpG island promoters. The same comment holds for XPG and XPG. Or Ddx19, which was not found here even though a plethora of DDX enzymes were identified. Of course, RNase H1 is not identified either... and only one subunit of the RNase H2 complex appears. The authors would do the field a favor by pointing these glaring absences. The authors should also comment on why these interactions were missed. Either their method is insufficient to identify these interactions (see previous paragraph), or the previous claims are simply unsupported (another possibility is that the factors are not expressed in the cells used here which is quite unlikely for many of them and easy to verify for the authors from proteomics and RNA-seq datasets). I personally favor the latter option, but the point is that this needs to be openly discussed. This may be uncomfortable, or feel distracting, for the authors. I can assure them that there is real value in pointing out that some of the many, often wild, associations made between various protein factors and R-loops could not be observed here, given the strengths and limitations of the method. Right now, the manuscript only points out those interactions that could be substantiated; this feels good and is safe – but it is a missed opportunity for the field to engage in much-needed self-correction.

In a similar manner, the authors studiously avoid comparing their data to previous proteomics datasets (ref 43, 44). While a detailed comparison may not be needed, major differences should be better pointed out. It was recently suggested that the affinity of S9.6 for ribosomal RNAs may have caused a large enrichment of ribosomal proteins in Ref 43 (Smolka et al., 2021 - PMID: 33830170). How do the datasets compare on that front? Again, there is value in pointing out flaws in prior work so others can avoid them and a correct interpretation can emerge out of the literature.

On a more focused point, I would request that the authors invest more effort in breaking down the dominant "RNA splicing"-related factors into more revealing sub-categories. In particular, breaking down these components between U1, U2 and other spliceosome components, splicing regulators (HNRNPs and SRSFs) and other components would be more informative. In Figure 2, it is odd that CSTF1/3 are categorized as "RNA splicing" and not "3'-end processing". A similar argument could be made for ALYREF. Why isn't POLR2A not grouped with RNAPII regulation instead of splicing? And what's the difference between "transcriptional regulation" and "RNAPII regulation"? Similarly, why are there two "RNA splicing" categories? Why does DDX41 not appear? Overall, this figure feels amateurish and needs to be improved for consistency and clarity.

The hard turn to Ddx41, stylistically, is a little jarring. Why were these proteins selected out of the screen? The transition is very abrupt and for the rest of the manuscript, the screen is an afterthought. This being said, the cellular characterization of DDX41 is for the most part fine – the data is well done using classic essays and the figures are clear. The S9.6 IF data should, however, simply not be used given the findings that such signals are largely artefactual due to rRNA recognition (see reference above). If the authors want to use this assay, they at least have to deliver all necessary controls (RNases T1 and III in addition to RNase H). I strongly urge them to not engage in this flawed approach. I will note that the use of the HBD-GFP as an R-loop imaging tool is so far getting "a pass" but it is not clear to me that it functions as advertised either...

The main concern with this part is the novelty. Ddx41 becomes the N+1 factor to be involved in antagonizing R-loops formation in a manner than is correlated with DNA damage. It is regrettable that the authors chose not to delve beyond correlations and tackle the key questions that underlie the work. Is the R-loop accumulation responsible for DSBs? For this, the authors should measure DSBs directly and show that they coincide (or not) with accumulated R-loops (and that both are RNase H1-suppressible). Is R-loop accumulation responsible for the slow replication fork velocity phenotype? It seems not, at least to me, since the slow replication phenotype appears global when only ~1,500 regions of accumulated R-loops were identified genome-wide. Simply stating that these phenotypes can be suppressed by active RNase H1 is unfortunately not sufficient to prove that the accumulation of promoter R-loops are causal of these phenotypes, as recently shown by Promonet and colleagues who carefully considered these issues (PMID: 32769985). Finally, it seems to me that the authors missed an opportunity to characterize the nascent transcription defects caused by Ddx41 depletion. An increased in MapR promoter signals is most likely due to an increase in RNAP II pausing. It would be worth investigating the broader effects on transcription using nascent transcription sequencing approaches such as TT-seq. I request that at the very least, authors discuss more thoughtfully their existing results and the pretty obvious disconnect between a pretty localized R-loop increase and a global replication effect. To make the story more appealing, a DSB mapping approach is strongly suggested.

In their biochemical characterization, why didn't the authors include three-stranded R-loop substrates in their assays (compared to bubble substrates)? This seems like a glaring omission. In the ATPase assay was the ssDNA overhang necessary for activity? This should be discussed. Overall, I find this portion relatively uninformative. Does the helicase have directionality? What is the k_{cat} for the ATPase activity, the helicase activity? Is it processive? None of these measurable parameters are shown. And finally, how does Ddx41 compare to other well-known enzymes?

Minor comments:

Line 266: "MCM complex was demonstrated..." that sentence is incomplete.

What cells were used for the RDProx screen? This is not clear.

Reviewer #2:

Remarks to the Author:

Mosler et al. use sensitive proximity proteomics to identify proteins associated with R-loops. R-loops have raised a lot of attention in recent years, and characterizing their roles and identifying the proteins involved in their regulation has become a hot topic. Specifically, the authors fused the RNA-DNA hybrid-binding domain (HBD) of RnaseH1 to APEX2, enriched biotinylated proteins from HEK293T cells and analyzed them by LC-MS/MS in SILAC conditions. A RNA-DNA-hybrid binding-deficient mutant harboring three point mutations in the HBD was used as specificity control. In technically well-controlled and reproducible experiments a total of 312 proteins were identified, including several known factors associated with R-loop regulation, but also many proteins that have not yet been linked to R-loop biology. The authors then focus on one of them, the DEAD box helicase DDX41, for which they provide evidence that it can unwind RNA-DNA hybrids in vitro. Depletion of DDX41 results in replication stress and DNA damage and at least some of this seems to come from increased R-loops. DDX41 mutations occur in AML and expression of disease variants of DDX41 in HSPCs seems to increase RNA-DNA-hybrid levels. Finally, the authors mapped hybrids after DDX41 depletion and also performed RNA sequencing, pointing to deregulated gene expression when DDX41 is missing.

The R-loop proximity proteomics provides a useful resource, although only for a single cell line and one condition. More critically, the characterization of DDX41, its role as R-loop processing enzyme and its relevance, seems incomplete and not yet fully convincing.

Main points:

- 1) For the proteomics part, the authors mention in their discussion two previous studies by Cristini et al. and Wang et al. (references 43 and 44) in which proteins associated with R-loops have been identified. It would be informative to compare the results and discuss the overlaps between these two studies and the current RDProx manuscript, also with regard to the advantages of RDProx such as measuring low abundant proteins and transient interactions.
- 2) The authors also mention in their discussion that RDProx is easily applicable for mapping R-loop-proximal proteins in different cell lines and species as well as upon different cellular perturbations. With this in mind, it is a pity that only one cell line and one condition, but no conditions of cell stress (e.g. to induce NFkappaB signaling, see below) and additional cell lines, were used in the manuscript.
- 3) Can it be excluded that the 1mM H₂O₂ treatment used for the proximity labeling induces R-loops? Controls or references to prior work excluding this possibility should be provided.
- 4) Can it be excluded that RNA-DNA hybrids are detected by the HBD, which are not R-loops? How about primers synthesized during DNA replication, for instance?
- 5) gH2AX levels, pRPAS33 and 53BP1 foci as well as replication fork slowing are used as proxy for R-loops and R-loop-induced DNA damage, but what is the evidence that they do not mark other types of replication stress and DNA damage, unrelated to R-loops? Unless they disappear after RnaseH1 expression in the specific conditions in which they are measured (in Figures 3d, 3e, 3f, S2b, S2c, 4e, 4f, S3a-c), whether or not they are caused by R-loops or other sources of replication stress is not clear.
- 6) On a related note, the rescue by RnaseH1 in Figure 3b is mild, with much of the gammaH2AX signal from the DDX41 knockdown being insensitive to RnaseH1. Does this imply that structures other than R-loops cause the problems in DDX41-depleted cells? The link to R-loops is not fully convincing.
- 7) Moreover, R-loops are induced by DNA double-strand breaks and play a role in DNA repair. Can the authors exclude such a scenario for DDX41 depleted cells?

- 8) The interpretation that the phosphorylation of RPA was mediated by ATR seems to be an overstatement, based on the results, because the reduction in the pRPA signal after ATR inhibition is rather moderate (Fig. S2d).
- 9) Figure 4a should contain higher magnifications of single cells, and images from the WKK mutant should be included.
- 10) The DRB rescue in Figure S3a is very mild, suggesting that a significant part of the signal may not stem from transcription-induced R-loops.
- 11) The S9.6 signal in IF is questionable, see Smolka et al., JCB 2021. The signal may stem from ribosomal RNA and other nucleic acid species and may not represent R-loops. The appropriate controls (RnaseH1 and Rnase T1 treatments) should be used for the conditions analyzed.
- 12) For the in vitro part, the authors write: "We found that the intrinsic ATPase activity of DDX41 was stimulated by RNA-DNA hybrids with a single-stranded DNA overhang (Figure 5c, Supplementary Figure 4b)". I could not find data in these two figure panels that support this interpretation.
- 13) Protein inputs should be shown for the experiments in Figure 5d and 5e. In particular for 5e such controls are critical, because the effect of the R525H mutation is mild.
- 14) The connection to TGFbeta, NOTCH and NFkappaB signaling is very vague and more work is required. For instance, does DDX41 bind to CGI promoters where R-loops accumulate? ChIP-qPCR or ChIP-seq would be needed to address this point. What about inducing TGFbeta, NOTCH or NFkappaB signaling and testing whether this leads to elevation of R-loops at target promoters in a manner that is modulated or antagonized by DDX41? Would this lead to replication-transcription conflicts and transcription-dependent DNA damage in DDX41-depleted and in DDX41-mutated (R525H, L237/P238T) cells?

Additional points:

- 1) Were the nucleotide excision repair proteins XPF and XPG, which the Cimprich lab had shown to process R-loops, identified in the RDProx approach?
- 2) The term RNA splicing appears twice in Fig. 2a.
- 3) A knockdown control for DDX39A is missing.
- 4) Controls for the DRB-induced transcriptional repression should be provided.
- 5) In Figure S2a the alignment of conditions in the left and right panels should be matched.
- 6) The serine residue that is phosphorylated by DNA damage kinases on H2AX is conventionally referred to as Ser139.

Reviewer #3:

Remarks to the Author:

In this paper, Mosler et al used a proximity biotinylation approach called RDProx to identify an R-loop proximal proteome. To this end, the authors fused the hybrid-binding domain of RNaseH to APEX2 proximal proteins were subsequently enriched using streptavidin beads and identified using SILAC based quantitative interaction proteomics. Follow up experiments revealed that the R-loop interacting protein DDX41 as a protein that opposes R-loop dependent genome instability. The authors further

identify a role for DDX41 in inflammatory signalling. In general terms, the work presented in this study is solid and worth reporting, but there are some major technical and conceptual flaws which severely dampen my enthusiasm for this study and which need to be addressed prior to publication

Major issues:

- First of all, based on their presented data (Figure 1 and 2), the authors identified a large number (~600) of putative R-loop interacting proteins. Many of these, however, are also known to interact with regular DNA or RNA structures. The question therefore is how specific these detected interactions really are. The authors need to provide biochemical evidence that the hybrid-binding domain of RNaseH exclusively interacts with R-loop structures but not with other types of nucleic acids such as DNA or RNA. Canonical DNA or (ds)RNA will be much more abundant in cells compared to R-loops so even of the affinity of the hybrid-binding domain of RNaseH is say 30 times higher for R-Loops compared to DNA or (ds)RNA, in vivo still a lot of the RNaseH-hybrid binding domain fused to APEX2 will interact with these nucleic acids due to the large access of DNA and RNA compared to R-Loops. Without such specificity controls the presented R-loop proteome is meaningless. The authors may also pursue RDProx experiments in WT versus DDX41 knockdown cells to see which of their putative R loop proteins are more enriched in DDX41 KD versus WT cells (since R-loops are more abundant in DDX41 KD cells).
- The functional and mechanistic connections between DDX41, R-loops, inflammatory gene expression and leukemia are anecdotal and mechanistically weak. For example, the authors hypothesize that R-Loop accumulation upon a loss of DDX41 might repel DNA methyltransferases or promote the recruitment of TET proteins to increase gene expression but this is purely speculative at this point. Intuitively, I would expect that accumulation of R-loops at promoters (observed upon DDX41 depletion) would result in reduced gene expression.

Minor comments:

- When first using the term RDProx the authors should specify the acronym
- Panels are missing in Figure 4A
- The volcano plot shown in Figure 1B is skewed, thus compromising outlier statistics. Set cut-offs are arbitrary

Point-by-point response to the Reviewers' comments

Mosler et al., "R-loop proximity proteomics identifies a role of DDX41 in transcription-associated genomic instability"

REVIEWER COMMENTS

Reviewer #1 (Remarks to the Author):

This manuscript describes an R-loop proteomics screen followed after a hard pivot by the characterization of the genome instability defects caused in human cells by deficiency in the Ddx41 helicase. The manuscript has its strengths and is generally well-presented and easy to read. It suffers, however, from significant shortcomings that are detailed below. Maybe the biggest flaw as things stand now is that the data is not particularly novel and is limited to a correlative analysis of the relationship between R-loops and phenomena of genome instability.

One shortcoming of the manuscript is that it doesn't acknowledge, and therefore oversells, the proteomics screen. The issue here is that the use of the HBD domain as an R-loop sensor guarantees that only a subset of R-loops are effectively targeted. These correspond primarily to short R-loops that form at paused promoters mostly corresponding to CpG islands. This reflects a recent proposal (PMID: 33411340) that simply takes into account the vast differences between in situ mapping approaches (typically RNase H1-based like MapR) and ex vivo mapping approaches (typically S9.6-based like DRIP). I request that the authors more specifically discuss this issue and highlight that the screen is most sensitive to a subset of R-loops mapping primarily to paused promoters. This matters much for making precise claims and also serves to explain the results of the screen. Promoter-proximal pausing is a well-established quality control step that permits the coordination of transcription with vital RNA processing complexes such as the spliceosome and RNA export machinery. Not surprisingly, these complexes are the top scoring complexes. Similarly, CpG island promoters are well-known to associate with replication origins (for instance PMID: 19360092) which could account for the observed links with replication proteins. Finally, the pause site are located right before the highly dynamic +1 nucleosome which could explain the identification of chromatin remodeling complexes. Overall, this goes to say that the regio-specificity inherent to the screen should be directly acknowledged and discussed better than it is now. The proteomics screen is interesting but given very short change in the manuscript. What shines brightly from the results is the absence of many factors hypothesized to play key roles in R-loop metabolism including many mentioned in the Introduction. For instance, SETX does not appear here. Nor do GADD45A and TET1 even though they were supposed to be recruited to R-loops at CpG island promoters. The same comment holds for XPG and XPG. Or Ddx19, which was not found here even though a plethora of DDX enzymes were identified. Of course, RNase H1 is not identified either... and only one subunit of the RNase H2 complex appears. The authors would do the field a favor by pointing these glaring absences. The authors should also comment on why these interactions were missed. Either their method is insufficient to identify these interactions (see previous paragraph), or the previous claims are simply unsupported (another possibility is that the factors are not expressed in the cells used here which is quite unlikely for many of them and easy to verify for the authors from proteomics and RNA-seq datasets). I personally favor the latter option, but the point is that this needs to be openly discussed. This may be uncomfortable, or feel distracting, for the authors. I can assure them that there is real value in pointing out that some of the many, often wild, associations made between various protein factors and R-loops could not be observed here, given the strengths and limitations of the method. Right now, the manuscript only points out those interactions that could be substantiated; this feels good and is safe – but it is a missed opportunity for the field to engage in much-needed self-correction.

We thank the Reviewer for raising these points.

We acknowledge that RDPprox is inherently biased to the subset of R-loops bound by RNaseH1 and discuss why some of the expected or previously reported proteins were not identified by adding the following paragraph in the discussion of the revised manuscript:

“RDProx relies on the HBD of RNaseH1 and is therefore inherently biased to the R-loops that are recognized and bound by RNaseH1. Recently, the existence of different classes of R-loops - promoter-paused R-loops and elongation-associated R-loops - that each display unique characteristics was proposed^{57,58}. Promoter-paused R-loops are short R-loops frequently forming during promoter-proximal pausing of RNA polymerase II^{57,58}. R-loop mapping approaches based on RNaseH1 showed an enrichment of RNaseH1 at promoter-proximal sites^{43,46}. It remains unclear whether and to which extent RNaseH1 binds to R-loops in other genomic regions. We therefore speculate that RDProx might be most sensitive in recovering proteins that associate with promoter-proximal R-loops. This could explain why some previously reported R-loop-associated proteins, such as the RNA/DNA helicase SETX and endonucleases XPG/XPF, were not identified in RDProx. For instance, SETX seems to primarily associate with R-loops at DSBs⁵⁹. On the other hand, we would expect XPG and XPF in proximity to transcription-associated R-loops but it might be that these proteins are recruited only in occasions when R-loops are processed into DSBs and when not any more bound by RNaseH1¹⁹. In addition, XPG and XPF are relatively low abundant in cells, which might preclude their identification by mass spectrometry. Similar might be true for RNase H2 complex, where we only identified RNase H2A but not the B and C subunit of the complex.”

DDX19

Under unperturbed conditions, DDX19 was shown to localize to the nuclear pore and to be recruited to R-loops only after DNA damage in an ATR/Chk1-dependent manner (Hodroj et al., 2017, EMBO J, PMID: 28314779).

RNaseH1

We do not expect to identify RNaseH1 with RDProx because the R-loops of which proximal proteome is determined will be primarily bound by exogenously expressed HBD fused with APEX2 and not with endogenous RNaseH1.

In a similar manner, the authors studiously avoid comparing their data to previous proteomics datasets (ref 43, 44). While a detailed comparison may not be needed, major differences should be better pointed out. It was recently suggested that the affinity of S9.6 for ribosomal RNAs may have caused a large enrichment of ribosomal proteins in Ref 43 (Smolka et al., 2021 - PMID: 33830170). How do the datasets compare on that front? Again, there is value in pointing out flaws in prior work so others can avoid them and a correct interpretation can emerge out of the literature.

We have compared RDProx with the dataset based on the immunoprecipitation with the S9.6 antibody (Cristini et al., 2018, PMID: 29742442). This comparison is shown now in Supplementary figure 1h in the manuscript. For unbiased inspection of the two different screens, we took the significantly enriched proteins in both datasets (for RDProx this amounted to 311 proteins, and for S9.6 IP to 469 proteins) and performed GO-BP term enrichment analysis. Indeed, as pointed out by the Reviewer the most significantly enriched terms in the S9.6 IP were “rRNA processing” and “ribosome biogenesis, whereas in the RDProx these terms did not show any enrichment.

Supplementary figure 1h: GO Term-Biological Process analysis of protein groups identified in S9.6 immunoprecipitation-based mass spectrometry screen (Cristini et al., 2018) and in this study by RDProx (Tier 1 proteins were used for the analysis). The 5 GO-BP terms with the highest $-\log_{10}$ adjusted p-value (Fisher's exact test, Bonferroni correction) are depicted.

Similar was true for *in vitro* pull down (Wang et al., 2018, PMID: 30108179), where the most enriched terms were "rRNA metabolic process" and "ribosome biogenesis" (Figure 1 in response letter). This was not true for the RDProx where the most significantly enriched terms were "mRNA processing" and "mRNA splicing", whereas "rRNA processing" and "ribosome biogenesis" did not show any enrichment at all.

Figure 1: Comparison of RDProx with *in vitro* pull down with R-loop-like substrate from Wang et al., 2018. The analysis was done as described in Supplementary figure 1h shown above.

We added the following paragraph in the discussion: "Recent reports showed that the S9.6 antibody in fixed human cells predominantly recognizes ribosomal RNA and not RNA-DNA hybrids⁵⁶. Indeed, unbiased inspection of the previously reported S9.6-based proteomics dataset by GO term enrichment analysis revealed "rRNA processing" and "ribosome biogenesis" as most significantly enriched terms (Supplementary figure 1f)".

On a more focused point, I would request that the authors invest more effort in breaking down the dominant "RNA splicing"-related factors into more revealing sub-categories. In particular, breaking down these components between U1, U2 and other spliceosome components, splicing regulators (HNRNPs and SRSFs) and other components would be more informative. In Figure 2, it is odd that CSTF1/3 are

categorized as “RNA splicing” and not “3'-end processing”. A similar argument could be made for ALYREF. Why isn't POLR2A not grouped with RNAPII regulation instead of splicing? And what's the difference between “transcriptional regulation” and “RNAPII regulation”? Similarly, why are there two “RNA splicing” categories? Why does DDX41 not appear? Overall, this figure feels amateurish and needs to be improved for consistency and clarity.

We thank the Reviewer for this suggestion. We modified Figure 2 to improve the representation of different groups of proteins that were found in the R-loop proximal proteome. We did not rely only on GO terms but also on manual annotation of proteins into different biological processes based on literature. In this way, we could break down general terms such as RNA splicing into U1 snRNP complex, U2 snRNP complex, U4/U6/U5 snRNP complexes, spliceosome regulation and spliceosome assembly. We think the new figure is more informative.

The hard turn to Ddx41, stylistically, is a little jarring. Why were these proteins selected out of the screen? The transition is very abrupt and for the rest of the manuscript, the screen is an afterthought.

We have now added a following sentence that makes the transition from the proteomics screen to characterization of DDX41 more intuitive:

“Dead box helicase 41 (DDX41) is a poorly characterized tumor suppressor and pathogenic variants in DDX41 cause familial MDS and AML, which prompted us to investigate its potential role in R-loop metabolism⁴¹.”

This being said, the cellular characterization of DDX41 is for the most part fine – the data is well done using classic essays and the figures are clear. The S9.6 IF data should, however, simply not be used given the findings that such signals are largely artefactual due to rRNA recognition (see reference above). If the authors want to use this assay, they at least have to deliver all necessary controls (RNases T1 and III in addition to RNase H). I strongly urge them to not engage in this flawed approach. I will note that the use of the HBD-GFP as an R-loop imaging tool is so far getting “a pass” but it is not clear to me that it functions as advertised either...

We thank the reviewer for pointing this out. Since experiments using overexpression of RNaseH1 or *in situ* digestion with RNaseH1 were challenging to perform in HSPCs in a reasonable time, we decided to remove these data from the manuscript. We only kept the data that shows that HSPCs upon depletion of DDX41 or expression of pathogenic DDX41 variants show accumulation of 53BP1 foci, pointing to DSBs and genomic instability. These data were moved to the last figure (Figure 6a, b) of the revised manuscript and follow the new BLISS data (Figure 5, Supplementary figure 5) demonstrating that DDX41 knockdown HCT116 cells accumulate DSBs in promoters.

The main concern with this part is the novelty. Ddx41 becomes the N+1 factor to be involved in antagonizing R-loops formation in a manner than is correlated with DNA damage. It is regrettable that the authors chose not to delve beyond correlations and tackle the key questions that underlie the work. Is the R-loop accumulation responsible for DSBs? For this, the authors should measure DSBs directly and show that they coincide (or not) with accumulated R-loops (and that both are RNase H1-suppressible).

We have now performed BLISS in 2 replicates and MapR in 3 replicates in HCT116 cells to map double strand breaks (DSBs) and R-loops, respectively, in the presence or absence of DDX41 (Figure 5, Supplementary figure 5). In addition, we performed greenCut&Run in 2 replicates to shows that GFP-DDX41 preferentially associates with promoters (Figure 4, Supplementary figure 4).

Knockdown of DDX41 in HCT116 cells (similar as in U2OS cells) led to an accumulation of R-loops in promoters (Figure 4, Supplementary figure 4, 5). In HCT116 cells, 7,275 R-loops showed a gain and 2 R-loops showed a loss in DDX41 KD cells (based on $FC > 1.5$). In U2OS cells, 6,810 R-loops showed a gain and 505 showed a loss after DDX41 KD (based on $FC > 2$). 30% (5,506 out of 18,811) of R-loops in promoter regions showed a gain ($FC > 2$) after DDX41 KD in U2OS cells, pointing to a prominent role of DDX41 in opposing R-loops in promoters of active genes. We used a fold change of 1.5 in HCT116 cells to determine R-loops gains because the distribution of the \log_2 normalized counts of siDDX41/siCtrl was more narrow than in U2OS cells (Figure 2 in response letter).

Figure 2: Distribution of \log_2 normalized counts for siDDX41/siCtrl is plotted for HCT116 and U2OS cells.

In wild type cells, we mapped 4,540 DNA fragility hotspots and in DDX41 KD cells, we mapped 8,381 DNA fragility hotspots using BLISS. We identified 1,642 promoter regions that showed increased fragility in DDX41 knockdown cells, and 53% of those promoters with DSB gains also displayed R-loops upon DDX41 loss (Figure 5e, f, Supplementary figure 5e). This suggests that a large proportion of sites with increased fragility upon DDX41 loss coincides with R-loops. To the best of our knowledge, this is the first study that reports mapping of DSBs and R-loops in the absence of the R-loop-resolving helicase, and that demonstrates that DNA fragility hotspots coincide with R-loops.

We did not try to perform BLISS in conditions of RNaseH1 overexpression. During the revision process, we noticed that overexpression of RNaseH1 already leads to increased phosphorylation of RPA on Serine 33, likely a consequence of hyper-resection of endogenous DSBs (Figure 3 in the response letter). In addition, in immunofluorescence analysis, we used for quantification only cells that express moderate levels of tagged RNaseH1 to exclude potentially negative effects of strong RNaseH1 overexpression on genome stability. This is not possible to do with BLISS where we would detect DNA fragility hotspots in total cell population. We noticed this problem with different levels of RNaseH1 expression even when selecting single clones. Therefore, we concluded that the interpretation of the BLISS data would be very challenging and not feasible in the revision process.

Figure 3: High content confocal microscopy of pRPA (ser33) in U2OS cells after overexpression of RNaseH1, DDX41 KD or RNaseH1 OE and DDX41 KD. Boxplot is displaying the 10th-90th percentile, black line indicates the median.

Is R-loop accumulation responsible for the slow replication fork velocity phenotype? It seems not, at least to me, since the slow replication phenotype appears global when only ~1,500 regions of accumulated R-loops were identified genome-wide. Simply stating that these phenotypes can be suppressed by active RNase H1 is unfortunately not sufficient to prove that the accumulation of promoter R-loops are causal of these phenotypes, as recently shown by Promonet and colleagues who carefully considered these issues (PMID: 32769985).

In the original submission, we determined the significant R-loop gains in U2OS cells using DiffBind (differential binding affinity analysis of ChIP-seq peak data), which is a gold standard for analyzing ChIP-seq datasets. DESeq analysis that is part of the DiffBind assumes that the majority of the population does not show a change. This was, however, not the case for MapR in wild type and DDX41 knockdown U2OS cells, and in particular, the population was completely skewed to positive fold changes in HCT116 cells (Figure 4d and Supplementary figure 5d). In the revised manuscript, we decided not to use anymore DiffBind analysis but to determine R-loop gains dependent on the fold changes as also done in other recent publications that determined R-loop gains and losses (PMID: 33986538, PMID: 32439635). To this end, a set of consensus regions was created using the intersection of peaks called per group (either in siDDX41 or siCtrl three replicates). Then the union of these two peak sets was used to quantify the signal present in the samples. Using R/Bioconductor packages the fold change for the consensus regions was calculated using the average normalized coverage (rpkm) of the regions. Normalization was based on the total amount of sequenced reads. We used a fold change >2 in U2OS cells to determine R-loops gains.

In U2OS cells, 6,810 R-loops showed a gain and 505 showed a loss after DDX41 KD. 30% (5,506 out of 18,811) of R-loops in promoter regions showed a gain in DDX41 knockdown cells, pointing to a prominent role of DDX41 in opposing R-loops in promoters of active genes. Similar to U2OS cells, loss of DDX41 in HCT116 cells led to a dramatic accumulation of R-loops in promoters of active genes (Supplementary figure 5b, c). From 15,177 MapR peaks identified in all 3 MapR replicate experiments in DDX41 knockdown cells, 7,275 showed a gain in R-loops (FC>1.5) (Supplementary figure 5d).

Therefore, we conclude that DDX41 opposes the accumulation of large proportion of R-loops found in promoter regions (30% based on FC>2), which could account for the decrease in replication fork speed and for the replication stress phenotype we observed in DDX41 KD U2OS cells. In addition, the reduction of replication speed does not necessarily occur just at forks that are directly affected by transcription-

replication conflicts but can be a global regulatory response that occurs as consequence of decrease in replication speed at a subset of forks that are obstructed by R-loops (PMID: 28714480, PMID: 19838172, PMID: 24636987, PMID: 21979917).

We added the following paragraph in the discussion: “We observed that DDX41 knockdown cells display slower replication fork progression and signatures of ATR-dependent signaling, suggesting that R-loops accumulated in DDX41 knockdown cells can obstruct progression of replication forks and lead to replication stress. We also found that DDX41 knockdown cells show signs of perturbed transcription initiation and elongation. It remains to be addressed whether R-loop accumulation leads to DSBs by interfering with replication and/or transcription machinery, and through which mechanisms and under which conditions these R-loops are processed into DSBs.”

Finally, it seems to me that the authors missed an opportunity to characterize the nascent transcription defects caused by Ddx41 depletion. An increase in MapR promoter signals is most likely due to an increase in RNAP II pausing. It would be worth investigating the broader effects on transcription using nascent transcription sequencing approaches such as TT-seq.

We thank the Reviewer for this suggestion. We monitored changes in nascent transcription using EU incorporation and immunofluorescence using antibodies against S2 and S5 phosphorylation of RBP1 CTD (Supplementary figure 4g, h).

The following paragraph is added in the results section: “Accumulation of R-loops at promoter regions was not accompanied by a global defect in nascent transcription based on 5-Ethynyl Uridine (EU) incorporation (Supplementary figure 4g). However, we observed a mild but significant decrease in serine 5- and an increase in serine 2-phosphorylation of the RBP1 C-terminal domain (CTD), suggesting that RNAPII initiation and possibly elongation are perturbed by accumulated R-loops in promoter regions (Supplementary figure 4h).”

We discuss now these findings and their relation to R-loop accumulation also in the discussion part of the manuscript (please see above).

I request that at the very least, authors discuss more thoughtfully their existing results and the pretty obvious disconnect between a pretty localized R-loop increase and a global replication effect. To make the story more appealing, a DSB mapping approach is strongly suggested.

Please see answers above.

In their biochemical characterization, why didn't the authors include three-stranded R-loop substrates in their assays (compared to bubble substrates)? This seems like a glaring omission. In the ATPase assay was the ssDNA overhang necessary for activity? This should be discussed. Overall, I find this portion relatively uninformative. Does the helicase have directionality? What is the k_{cat} for the ATPase activity, the helicase activity? Is it processive? None of these measurable parameters are shown. And finally, how does Ddx41 compare to other well-known enzymes?

We assayed the *in vitro* binding affinity of purified full length DDX41 to a variety of oligonucleotides by fluorescence polarization assay. DDX41 displayed the strongest affinity towards RNA-DNA hybrids with a K_d of $2.5\mu\text{M} \pm 1.4\mu\text{M}$. To our knowledge, none of the prior “R-loop helicase” publications have provided information about the binding affinity or association constant (PMID: 29742442, PMID: 28790157, PMID: 31267554, PMID: 32439635, PMID: 30108179). Unfortunately, this makes it impossible to compare the binding affinity of DDX41 to other potential DEAD/H-box R-loop helicases.

However, we monitored now the binding of the hybrid-binding domain of RNaseH1 to RNA-DNA hybrid substrates and found that it associates with a K_d of 0.19 μM . These results suggest that DDX41 binds 10-fold weaker to RNA-DNA hybrids compared to the HBD of RnaseH1. We discuss this in the results section of the revised manuscript.

The setup of our fluorescence-based unwinding assay did not allow us to test the R-loop substrate. In order to test the displacement of the RNA moiety from the substrate, the quencher-fluorophore pair needs to be positioned on the RNA-DNA hybrid moiety. This would require to place the quencher/fluorophore within the DNA sequence, which was not possible in our unwinding assay.

The ssDNA overhang was not necessary for ATP hydrolysis. We observed similar activity towards blunt-ended RNA-DNA hybrid substrates. Since the substrate with the ssDNA overhang is a bit closer to the *in vivo* situation, we decided to go ahead with the overhang substrate. With the ATPase assay, we found that the k_{cat} for ATP hydrolysis by DDX41 was in the range of $1.43\text{-}2.51 \times 10^{-5}$ 1/s. The lack of this information in other DEAD/H-box helicase studies towards RNA-DNA hybrids again impedes a comparison. We now added the measurable parameters in the manuscript text and figures.

Minor comments:

Line 266: "MCM complex was demonstrated..." that sentence is incomplete.

We fixed that sentence: "The MCM complex was demonstrated to possess the RNA-DNA helicase activity *in vitro* and is therefore potentially involved in the removal of R-loops in S phase".

What cells were used for the RDPprox screen? This is not clear.

The RDPprox screen was performed in HEK293T cells. This information is present in the results describing Figure 1 and in the figure legend.

Reviewer #2 (Remarks to the Author):

Mosler et al. use sensitive proximity proteomics to identify proteins associated with R-loops. R-loops have raised a lot of attention in recent years, and characterizing their roles and identifying the proteins involved in their regulation has become a hot topic. Specifically, the authors fused the RNA-DNA hybrid-binding domain (HBD) of RnaseH1 to APEX2, enriched biotinylated proteins from HEK293T cells and analyzed them by LC-MS/MS in SILAC conditions. A RNA-DNA-hybrid binding-deficient mutant harboring three point mutations in the HBD was used as specificity control. In technically well-controlled and reproducible experiments a total of 312 proteins were identified, including several known factors associated with R-loop regulation, but also many proteins that have not yet been linked to R-loop biology. The authors then focus on one of them, the DEAD box helicase DDX41, for which they provide evidence that it can unwind RNA-DNA hybrids *in vitro*. Depletion of DDX41 results in replication stress and DNA damage and at least some of this seems to come from increased R-loops. DDX41 mutations occur in AML and expression of disease variants of DDX41 in HSPCs seems to increase RNA-DNA-hybrid levels. Finally, the authors mapped hybrids after DDX41 depletion and also performed RNA sequencing, pointing to deregulated gene expression when DDX41 is missing.

The R-loop proximity proteomics provides a useful resource, although only for a single cell line and one condition. More critically, the characterization of DDX41, its role as R-loop processing enzyme and its relevance, seems incomplete and not yet fully convincing.

Main points:

For the proteomics part, the authors mention in their discussion two previous studies by Cristini et al. and Wang et al. (references 43 and 44) in which proteins associated with R-loops have been identified. It would be informative to compare the results and discuss the overlaps between these two studies and the current RDPprox manuscript, also with regard to the advantages of RDPprox such as measuring low abundant proteins and transient interactions.

We have compared RDPprox with the dataset based on the immunoprecipitation with the S9.6 antibody (Cristini et al., 2018, PMID: 29742442). This comparison is shown in Supplementary Figure 1h copied above. For unbiased inspection of the two different screens, we took the significantly enriched proteins in both datasets (for RDPprox this amounted to 311 proteins, and for S9.6 IP to 469 proteins) and performed GO-BP term enrichment analysis. The most significantly enriched terms in the S9.6 IP were “rRNA processing” and “ribosome biogenesis, whereas in the RDPprox these terms did not show any enrichment.

Similar was true for *in vitro* pull down (Wang et al., 2018, PMID: 30108179), where the most enriched terms were “rRNA metabolic process” and “ribosome biogenesis” (Figure 1 in response letter). This was not true for the RDPprox where the most significantly enriched terms were “mRNA processing” and “mRNA splicing”, whereas “rRNA processing” and “ribosome biogenesis” did not show any enrichment at all.

We added the following paragraph in the discussion: “Recent reports showed that the S9.6 antibody in fixed human cells predominantly recognizes ribosomal RNA and not RNA-DNA hybrids⁵⁶. Indeed, unbiased inspection of the previously reported S9.6-based proteomics dataset by GO term enrichment analysis revealed “rRNA processing” and “ribosome biogenesis” as most significantly enriched terms (Supplementary figure 1f)”.

We do not have experimental evidence that proximity proteomics performs better in identifying low abundant proteins compared to classical affinity proteomics approaches. Therefore, we have rephrased the sentence that rather than being superb in identifying low abundant proteins, proximity proteomics

has the advantage when it comes to identifying chromatin-associated proteins that are difficult to solubilize since cell lysis can be done in 1% SDS in combination with sonication.

The text reads now as follows:

“The advantages of RDProx are manifold: (1) Labeling of R-loop-proximal proteins is performed *in vivo*, which ensures that R-loops, chromatin and cellular compartments remain intact; (2) Chromatin-associated proteins that are difficult to solubilize are amenable to the analysis; (3) Low affinity and transient interactions are detected. RDProx is easily applicable for mapping R-loop-proximal proteins in different cell lines and species as well as upon different cellular perturbations.”

Indeed, we used the data from Kustatscher et al. “Proteomics of a fuzzy organelle: interphase chromatin” to show that proteins identified in RDProx are more likely to be associated with chromatin compared to proteins identified using S9.6 IP (Figure 4 in response letter).

Figure 4: Probability scores for interphase chromatin presence were obtained from Kustatscher et al. for Tier1 proteins in RDProx and proteins identified in S9.6 IP (Cristini et al, 2018)

The authors also mention in their discussion that RDProx is easily applicable for mapping R-loop-proximal proteins in different cell lines and species as well as upon different cellular perturbations. With this in mind, it is a pity that only one cell line and one condition, but no conditions of cell stress (e.g. to induce NFkappaB signaling, see below) and additional cell lines, were used in the manuscript.

We performed RDProx upon knockdown of DDX41. We found that knockdown of DDX41 leads to increased proximity of 26 proteins (labeled in green) and decreased proximity of 65 proteins (labeled in blue) compared to wild type cells (FDR<5%) (Figure 5 in response letter). Proteins that showed more association with R-loops after DDX41 knockdown did not show an enrichment for any specific biological process. Interestingly, we found that proteins involved in “regulation of mRNA stability” and “mRNA splicing” showed less association with R-loops after DDX41. This suggest that spliceosome is displaced from R-loops that accumulate in promoter regions after DDX41 knockdown. Further experiments are needed to substantiate these findings and to understand the interplay between R-loop accumulation and spliceosome displacement. It is possible that spliceosome displacement is a consequence of RNAPII stalling at accumulated R-loops. Spliceosome displacement was previously shown to occur in response to RNAPII stalling after encountering transcription-blocking lesions induced by UV light (PMID: 26106861).

In addition, we performed RDProx-Western blot experiment in U2OS cells to confirm that proximity of DDX41 and AQR to R-loops is reduced when R-loops are suppressed by overexpression of RNaseH1 under the control of a doxycycline-inducible promoter (Supplementary figure 2a).

Figure 5: Volcano plot of protein groups identified by RDPprox in three biological replicates. Mean \log_2 ratios of all replicates between HBD siDDX41/HBD siCtr are plotted against the $-\log_{10}$ FDR. The FDR and enrichment were calculated using Limma. Significantly enriched proteins are highlighted in blue and green ($FDR < 0.05$) (left). GO-BP enrichment analysis of proteins that are found less proximal to HBD in absence of DDX41 (right).

Can it be excluded that the 1mM H₂O₂ treatment used for the proximity labeling induces R-loops? Controls or references to prior work excluding this possibility should be provided.

We confirmed that the treatment with H₂O₂ used exactly as for induction of biotinylation in RDPprox (1 mM, 2 min) does not lead to an increase of R-loop levels detectable by dot blot assay based on S9.6 (Figure 6 in response letter).

Figure 6: R-loop levels were examined in three replicate experiments using S9.6-based dot blot. Half of the sample was treated with RNaseH to control for the signal specificity. The gDNA was spotted in different concentrations and the membrane was probed with the S9.6 antibody before being stripped and consequently probed with dsDNA antibody as loading control.

Can it be excluded that RNA-DNA hybrids are detected by the HBD, which are not R-loops? How about primers synthesized during DNA replication, for instance?

Early publications that aimed to reconstitute mammalian Okazaki fragment processing showed that RNase H1 could remove primers *in vitro* (PMID: 7524089, PMID: 9482870). Recent studies provide no evidence that RNase H1 can remove primers during DNA replication *in vivo*. RNase H2- and Exo1-mediated exonucleolytic digestion and the Dna2- and Fen1-mediated flap cleavage are predominant mechanisms utilized by eukaryotic cells to remove RNA-DNA primers during replication (PMID: 28159842, PMID: 30541106). We also did not observe any cell cycle defects upon overexpression of RNase H1, arguing against that RNaseH1 plays a role in processing Okazaki fragments *in vivo*.

gH2AX levels, pRPAS33 and 53BP1 foci as well as replication fork slowing are used as proxy for R-loops and R-loop-induced DNA damage, but what is the evidence that they do not mark other types of

replication stress and DNA damage, unrelated to R-loops? Unless they disappear after RnaseH1 expression in the specific conditions in which they are measured (in Figures 3d, 3e, 3f, S2b, S2c, 4e, 4f, S3a-c), whether or not they are caused by R-loops or other sources of replication stress is not clear.

We found that DDX41 knockdown results in increased γ H2AX, 53BP1 and pRPA (S33) using immunofluorescence. We showed that doxycycline-inducible overexpression of RNaseH1 rescues the phenotype of DDX41 knockdown on γ H2AX measured by IF and DSBs measured by neutral comet assay. We now also show that overexpression of RNase H1 rescues the increased formation of 53BP1 foci in DDX41 knockdown cells (Supplementary figure 2c). We could not rescue the effect of DDX41 knockdown on pRPA (S33). On the contrary, RNase H1 overexpression in DDX41 knockdown cells increased pRPA intensity. In addition, overexpression of RNase H1 alone in wild type cells leads to increased pRPA intensity (Figure 2 in response letter).

We conclude that overexpression of RNase H1 and removal of RNA-DNA hybrids at endogenous DSBs leads to hyper-resection and increased recruitment of RPA, a phenotype that is even more pronounced in DDX41 knockdown cells. Indeed, in line with our results it was reported that overexpression of RNase H1 fosters resection at DSBs (PMID: 27881299, PMID: 33837288).

On a related note, the rescue by RnaseH1 in Figure 3b is mild, with much of the gammaH2AX signal from the DDX41 knockdown being insensitive to RnaseH1. Does this imply that structures other than R-loops cause the problems in DDX41-depleted cells? The link to R-loops is not fully convincing.

In addition to γ H2AX, we also showed that OE of RNase H1 can rescue the effect of DDX41 KD on DSBs measured by neutral comet assay (Figure 2c) and on 53BP1 foci formation (Supplementary figure S2d). As shown in Figure 2 in response letter, overexpression of RNase H1 already leads to increased phosphorylation of RPA on Serine 33, likely as consequence of hyper-resection of endogenous DSBs (Figure 3 in the response letter). These hyper-resected sites might also be a source of γ H2AX signal that could not be rescued in DDX41 KD cells upon OE of RNase H1.

To strengthen the findings that R-loop accumulation in DDX41 KD cells leads to genomic instability, we now performed BLISS and MapR in HCT116 cells to map double strand breaks (DSBs) and R-loops, respectively, in the presence or absence of DDX41 (Figure 5, Supplementary figure 5). Knockdown of DDX41 in HCT116 cells (similar as in U2OS cells) led to an accumulation of R-loops in promoters (Figure 4, Supplementary figure 4, 5). In HCT116 cells, 7,275 R-loops showed a gain and 2 R-loops showed a loss in DDX41 KD cells (based on $FC > 1.5$). In U2OS cells, 6,810 R-loops showed a gain and 505 showed a loss after DDX41 KD (based on $FC > 2$). 30% (5,506 out of 18,811) of R-loops in promoter regions showed a gain ($FC > 2$) after DDX41 KD, pointing to a prominent role of DDX41 in opposing R-loops in promoters of active genes. We used a fold change of 1.5 in HCT116 cells to determine R-loops gains because the distribution of the \log_2 normalized counts of siDDX41/siCtr was much tighter than in U2OS cells (Figure 2 in response letter). In wild type cells, we mapped 4,540 DNA fragility hotspots and in DDX41 KD cells, we mapped 8,381 DNA fragility hotspots using BLISS. We identified 1,642 promoter regions that showed increased fragility in DDX41 knockdown cells, and 53% of those promoters with DSB gains also displayed R-loops upon DDX41 loss (Figure 5e, f, Supplementary figure 5e). This suggest that a large proportion of DNA fragile sites in promoters coincides with R-loops in DDX41 knockdown cells. To the best of our knowledge, this is the first study that reports mapping of DSBs and

R-loops in the absence of the R-loop-resolving helicase, and that demonstrates that sites of R-loop accumulation coincide with DNA fragility hotspots.

Moreover, R-loops are induced by DNA double-strand breaks and play a role in DNA repair. Can the authors exclude such a scenario for DDX41 depleted cells?

RNA-DNA hybrids have been shown to play a role in DSB repair by homologous recombination (HR) (PMID: 27881299, PMID: 33837288). We monitored DSB repair by HR in DDX41 knockdown cells and did not see a significant effect of the DDX41 knockdown on this pathway (Figure 7 in response letter). Therefore, we conclude that DDX41 does not play a role in repair of DSBs.

Figure 7: Efficiency of homology-directed repair in DDX41 knockdown cells. U2OS cells were transfected with siRNA against DDX41 or CtIP (as positive control) and the efficiency of I-SceI-induced 48h-post transfection was measured using a traffic light reporter assay.

The interpretation that the phosphorylation of RPA was mediated by ATR seems to be an overstatement, based on the results, because the reduction in the pRPA signal after ATR inhibition is rather moderate (Fig. S2d).

We rephrased this sentence in the revised manuscript: “Moreover, the phosphorylation of RPA was to some extent mediated by ATR, since inhibition of ATR with VE-821 significantly reduced the increase in pRPA (S33) intensity after DDX41 knockdown (Supplementary figure 2f)”.

Figure 4a should contain higher magnifications of single cells, and images from the WKK mutant should be included.

We have now added a new panel with higher magnifications of single cells as well as the images from the WKK mutant in Supplementary Figure 3a.

The DRB rescue in Figure S3a is very mild, suggesting that a significant part of the signal may not stem from transcription-induced R-loops.

We do not expect that DRB can completely rescue the effects of DDX41 knockdown on R-loop accumulation. DRB inhibits the elongation of paused RNAPII and RNAPII pausing has been suggested to induce R-loop accumulation. Indeed, it has been shown, using R-ChIP, that treatment of cells for 2 hours with DRB (condition which we used) results in some R-loop accumulation (PMID: 29104020). We had similar observations upon treatment of cells with the transcription inhibitor Actinomycin D that intercalates into the DNA. Actinomycin-treatment resulted in complete loss of R-loops in promoter regions measured by MapR but led to the accumulation of R-loops in downstream regions probably reflecting the sites of stalled RNAPII. Therefore, in our opinion the most relevant is to investigate how different R-loops in the genome are affected in response to DDX41 knockdown. To this end, we performed genome-wide analysis of R-loops in U2OS and HCT116 cells upon DDX41 knockdown and

this data clearly demonstrates that loss of DDX41 leads to an accumulation of R-loops in promoter regions. Comparison of MapR with RNA-seq (in U2OS cells) and GRO-seq (published dataset in HCT116 cells) shows that accumulation of R-loops upon DDX41 loss is dependent on transcription. In addition, we used greenCut&Run to map chromatin regions bound by DDX41 and found that binding of DDX41 to promoter regions positively correlates with gene expression. Combined, these data show that DDX41 opposes R-loop accumulation in promoter regions of active genes.

The S9.6 signal in IF is questionable, see Smolka et al., JCB 2021. The signal may stem from ribosomal RNA and other nucleic acid species and may not represent R-loops. The appropriate controls (RNaseH1 and RNase T1 treatments) should be used for the conditions analyzed.

Since experiments using overexpression of RNaseH1 or *in situ* digestion with RNaseH1 were challenging to do in HSPCs in a reasonable time, we decide to remove these data from the manuscript. We only kept the data that shows that HSPCs upon depletion of DDX41 or expression of pathogenic DDX41 variants show accumulation of 53BP1 foci, pointing to DSB and genomic instability. These data were moved to the last figure (Figure 6a, b) and follow the new BLISS data (Figure 5, Supplementary figure 5) demonstrating that DDX41 knockdown HCT116 cells accumulate DSBs in promoters.

For the *in vitro* part, the authors write: “We found that the intrinsic ATPase activity of DDX41 was stimulated by RNA-DNA hybrids with a single-stranded DNA overhang (Figure 5c, Supplementary Figure 4b)”. I could not find data in these two figure panels that support this interpretation.

We rephrased the sentence as follows: We employed an ATPase assay to determine whether the ATPase domain of DDX41 hydrolyzes ATP when encountering an RNA-DNA hybrid substrate. We found that that ATP hydrolysis by DDX41 was stimulated by RNA-DNA hybrids with a single-stranded DNA overhang (Supplementary figure 3f, g).

With the ATPase assay, we found that the *k_{cat}* for ATP hydrolysis by DDX41 was in the range of 1.43-2.51*10⁻⁵ 1/s. We also added this information in the figure.

Protein inputs should be shown for the experiments in Figure 5d and 5e. In particular for 5e such controls are critical, because the effect of the R525H mutation is mild.

We have now added a Coomassie stained gel that shows protein inputs used in the experiment in Supplementary figure 3h.

The connection to TGFbeta, NOTCH and NFkappaB signaling is very vague and more work is required. For instance, does DDX41 bind to CGI promoters where R-loops accumulate? CHIP-qPCR or CHIP-seq would be needed to address this point. What about inducing TGFbeta, NOTCH or NFkappaB signaling and testing whether this leads to elevation of R-loops at target promoters in a manner that is modulated or antagonized by DDX41? Would this lead to replication-transcription conflicts and transcription-dependent DNA damage in DDX41-depleted and in DDX41-mutated (R525H, L237/P238T) cells?

To test whether DDX41 can associate with chromatin *in vivo*, we generated U2OS cells that express GFP-tagged DDX41 under a doxycycline inducible promoter. We confirmed that these cells show pan-nuclear DDX41 staining that mirrored the localization of the endogenous DDX41 (Supplementary figure 4a). We used a GFP nanobody to target micrococcal nuclease to chromatin regions bound by DDX41 using greenCUT&RUN^{44,45}. We detected 19,327 DDX41 peaks in 2 biological replicate experiments, among which 6,363 were consistently found in both experiments (Supplementary table 2). Interestingly, DDX41 displayed preference to bind promoters with 41% of DDX41 peaks mapping to promoter regions

(TSS +/- 3kb) (Figure 4a, b). We identified 6,441 promoters bound by DDX41, and comparison of DDX41 binding sites with RNA-sequencing from U2OS cells revealed that DDX41 association with promoters depends on gene expression levels with association being stronger at highly expressed genes (Supplementary figure 4b). Nearly 40% of promoters that displayed accumulation of R-loops also associated with GFP-tagged DDX41 (Figure 4 h, Supplementary figure 4f). This is likely an underestimation since CUT&RUN was performed under very mild crosslinking conditions, and DDX41 loosely associates with chromatin.

We have toned down the parts on TGF β and NOTCH signaling in the revised manuscript by removing it from the abstract and introduction. In the results section, we show the Reactome pathway over-representation analysis of genes with accumulated R-loops that revealed chromatin organization, NOTCH and TGF β signaling among most significantly enriched terms (Figure 4g). In the discussion section, we added the following sentence to relate to the result shown in Figure 4g: "Genes that show R-loop accumulation are enriched for pathways frequently altered in AML such as chromatin organization, RUNX1 interactions as well as NOTCH and TGF β signaling suggesting that DDX41 loss results in dysregulated transcription and aberrant cellular signaling through those pathways."

Additional points:

Were the nucleotide excision repair proteins XPF and XPG, which the Cimprich lab had shown to process R-loops, identified in the RDPprox approach?

We did not identify XPF and XPG in the RDPprox. We discuss the shortcomings of the screen as follows: "RDPprox relies on the HBD of RNaseH1 and is therefore inherently biased to the R-loops that are recognized and bound by RNaseH1. Recently, the existence of different classes of R-loops - promoter-paused R-loops and elongation-associated R-loops - that each display unique characteristics was proposed. Promoter-paused R-loops are short R-loops frequently forming during promoter-proximal pausing of RNA polymerase II^{57,58}. R-loop mapping approaches based on RNaseH1 showed an enrichment of RNaseH1 at promoter-proximal sites^{43,46}. It remains unclear whether and to which extent RNaseH1 binds to R-loops in other genomic regions. We therefore speculate that RDPprox might be most sensitive in recovering proteins that associate with promoter-proximal R-loops. This could explain why some previously reported R-loop-associated proteins, such as the RNA/DNA helicase SETX and endonucleases XPG/XPF, were not identified in RDPprox. For instance, SETX seems to primarily associate with R-loops at DSBs⁵⁹. On the other hand, we would expect XPG and XPF in proximity to transcription-associated R-loops but it might be that these proteins are recruited only in occasions when R-loops are processed into DSBs and when not any more bound by RNase H1¹⁹. In addition, XPG and XPF are relatively low abundant in cells, which might preclude their identification by mass spectrometry. Similar might be true for RNase H2 complex, where we only identified RNase H2A but not the B and C subunit of the complex".

The term RNA splicing appears twice in Fig. 2a.

We modified this figure to better reflect groups of proteins in the R-loop proximal proteome. We did not rely only on GO terms but also on manual annotation of proteins into different biological processes based on literature. In this way, we could break down general terms such as RNA splicing into U1 snRNP

complex, U2 snRNP complex, U4/U6/U5 snRNP complexes, spliceosome regulation and spliceosome assembly. We think the new figure is more informative.

A knockdown control for DDX39A is missing.

We have now provided this control by performing a qPCR for DDX39A (Supplementary figure 2b).

Controls for the DRB-induced transcriptional repression should be provided.

We have now provided these controls in Supplementary figure 4g, h. We show that DRB treatment dramatically decreases EU incorporation and serine 2 RBP1 CTD phosphorylation.

In Figure S2a the alignment of conditions in the left and right panels should be matched.

We have corrected that.

The serine residue that is phosphorylated by DNA damage kinases on H2AX is conventionally referred to as Ser139.

We have corrected that.

Reviewer #3 (Remarks to the Author):

In this paper, Mosler et al used a proximity biotinylation approach called RDProx to identify an R-loop proximal proteome. To this end, the authors fused the hybrid-binding domain of RNaseH to APEX2 proximal proteins were subsequently enriched using streptavidin beads and identified using SILAC based quantitative interaction proteomics. Follow up experiments revealed that the R-loop interacting protein DDX41 as a protein that opposes R-loop dependent genome instability. The authors further identify a role for DDX41 in inflammatory signalling. In general terms, the work presented in this study is solid and worth reporting, but there are some major technical and conceptual flaws which severely dampen my enthusiasm for this study and which need to be addressed prior to publication

Major issues:

First of all, based on their presented data (Figure 1 and 2), the authors identified a large number (~600) of putative R-loop interacting proteins. Many of these, however, are also known to interact with regular DNA or RNA structures. The question therefore is how specific these detected interactions really are. The authors need to provide biochemical evidence that the hybrid-binding domain of RNaseH exclusively interacts with R-loop structures but not with other types of nucleic acids such as DNA or RNA. Canonical DNA or (ds)RNA will be much more abundant in cells compared to R-loops so even of the affinity of the hybrid-binding domain of RNaseH is say 30 times higher for R-Loops compared to DNA or (ds)RNA, *in vivo* still a lot of the RNaseH-hybrid binding domain fused to APEX2 will interact with these nucleic acids due to the large access of DNA and RNA compared to R-Loops. Without such specificity controls the presented R-loop proteome is meaningless.

RNase H1 acts as a monomer and harbors an N-terminal hybrid binding domain (HBD) and a C-terminal catalytic domain. *In vitro*, the HBD (residues 27-76) displays at least 25-fold higher affinity for RNA-DNA hybrids as compared to dsRNA (PMID: 18337749, PMID: 17964265). Recombinant GFP-tagged, catalytically inactive (D210N) human RNase H1 was recently reported as a sensitive and specific tool for *in situ* imaging of RNA-DNA hybrids in fixed cells (PMID: 3423228). We have added this information in the introduction since this is relevant for RDProx.

We confirmed now preferential binding behavior of the HBD for RNA-DNA hybrids *in vitro* using purified domains, HBD and HBD-WKK, and different nucleic acid substrates using fluorescence polarization and EMSA. This confirmed the preferential affinity of HBD for RNA-DNA hybrids and that the WKK mutant displays dramatically reduced affinity for hybrids (Supplementary Figure 1c, d).

We also performed RDProx-Western blot to confirm that proximity of DDX41 and AQR to R-loops *in vivo* is reduced when R-loops are suppressed by overexpression of RNaseH1 under the control of a doxycycline-inducible promoter further validating RDProx specificity to map the R-loop-proximal proteins (Supplementary figure 2a).

The authors may also pursue RDProx experiments in WT versus DDX41 knockdown cells to see which of their putative R loop proteins are more enriched in DDX41 KD versus WT cells (since R-loops are more abundant in DDX41 KD cells).

We have performed this experiment, and found that knockdown of DDX41 leads to increased proximity of 26 proteins (labeled in green) and decreased proximity of 65 proteins (labeled in blue) compared to wild type cells (FDR<5%) (Figure 5 in response letter, please see above). Proteins that showed more association with R-loops after DDX41 knockdown did not show an enrichment for any specific biological process. Interestingly, we found that proteins involved in “regulation of mRNA stability” and “mRNA

splicing” showed less association with R-loops after DDX41. This suggest that spliceosome is displaced from R-loops that accumulate in promoter regions after DDX41 knockdown. Further experiments are needed to substantiate these findings and to understand the interplay between R-loop accumulation and spliceosome displacement. It is possible that spliceosome displacement is a consequence of RNAPII stalling at accumulated R-loops. Spliceosome displacement was previously shown to occur in response to RNAPII stalling after encountering transcription-blocking lesions induced by UV light (PMID: 26106861). Known R-loop-associating proteins did not show any significant change upon knockdown of DDX41 (labeled in black).

In addition, we performed RDPProx-Western blot experiment in U2OS cells to confirm that proximity of DDX41 and AQR to R-loops is reduced when R-loops are suppressed by overexpression of RNaseH1 under the control of a doxycycline-inducible promoter (Supplementary figure 2a).

The functional and mechanistic connections between DDX41, R-loops, inflammatory gene expression and leukemia are anecdotal and mechanistically weak. For example, the authors hypothesize that R-Loop accumulation upon a loss of DDX41 might repel DNA methyltransferases or promote the recruitment of TET proteins to increase gene expression but this is purely speculative at this point. Intuitively, I would expect that accumulation of R-loops at promoters (observed upon DDX41 depletion) would result in reduced gene expression.

In the revised manuscript we have added a significant amount of new data that strengthens our findings that DDX41 opposes R-loops in promoter regions of active genes. We performed MapR to show that also in HCT116 cells (previously only in U2OS cells), loss of DDX41 results in R-loop accumulation in promoter regions (Figure 5, Supplementary figure 5). We employed BLISS to map DNA fragility hotspots in HCT116 wild type and DDX41 KD cells, which demonstrated that upon loss of DDX41 leads to a higher number of DNA fragility hotspots in promoter regions. More than 50% of these sites that showed increased fragility in DDX41 KD cells coincided with R-loops in DDX41 KD cells. Moreover, we have performed CUT&RUN to demonstrate that DDX41 preferentially binds to promoter regions on chromatin (Figure 4, Supplementary figure 4).

We have toned down the parts on AML and TGF β and NOTCH signaling in the revised manuscript by removing it from the abstract and introduction. In the results section, we show the Reactome pathway over-representation analysis of genes with accumulated R-loops that revealed chromatin organization, NOTCH and TGF β signaling (Figure 4g). In the discussion section, we added the following sentence to relate to the result shown in Figure 4g: “Genes that show R-loop accumulation are enriched for pathways frequently altered in AML such as chromatin organization, RUNX1 interactions as well as NOTCH and TGF β signaling suggesting that DDX41 loss results in dysregulated transcription and aberrant cellular signaling through those pathways.”

Minor comments:

When first using the term RDPProx the authors should specify the acronym

We have done this: “To gain insights into protein-based mechanisms that regulate R-loop homeostasis, we probed the R-loop-proximal protein networks using RNA-DNA proximity proteomics (RDPProx).”

Panels are missing in Figure 4A

We have now also provided images with the WKK mutant in Supplementary figure 3a.

The volcano plot shown in Figure 1B is skewed, thus compromising outlier statistics. Set cut-offs are arbitrary

The RDProx was performed in three biological replicate experiments and SILAC was used for robust relative quantification of protein groups proximal to HBD compared to HBD-WKK. We determined the false discovery rate using LIMMA (PMID: 16646809, PMID: 25821719), which is a standard in the proteomics field. To determine significantly enriched protein groups in APEX2-HBD vs APEX2-HBD-WKK we used an FDR<1% and HBD/WKK fold change >2 (Tier 2) or >4 (Tier 1). The skewing of the volcano plot must be a biological effect since HBD associates with R-loops whereas WKK does not and therefore HBD seems to be in proximity of a larger number of proteins. It is true that cut-offs based on FC>2 (Tier 2) and >4 (Tier 1) are somewhat arbitrary. However, in our experience it is helpful to make such cut-offs in addition to the FDR cut-off in order to easily distinguish proteins that show strong enrichment compared to the ones that show moderate enrichment (although both being significantly enriched based on FDR<1%). We are confident that this will be useful and convenient when selecting proteins from RDProx for future functional studies for their role in R-loop regulation.

Reviewers' Comments:

Reviewer #1:

Remarks to the Author:

The authors have answered my requests and the manuscript has been significantly improved as a result of this and new data. I look forward to the seeing the work in print.

Reviewer #2:

Remarks to the Author:

The authors significantly revised and improved the manuscript. The newly added DSB mapping by BLISS corroborates the conclusion on genomic instability in DDX41-deficient cells and links it to promoter regions of active genes, where also R-loops are found. The data and analyses are overall of high quality. While the new analyses strengthen the correlation between R-loops and DSBs, about half of the promoters with DSBs after DDX41 loss do not seem to display R-loops, and, more importantly, causality between elevated R-loops and DSB induction is not mandatory from the correlative analyses. This should be reflected a bit better in the manuscript text (see below).

Specific recommendations:

- 1) In the abstract, line 37, please consider including "may" before "contribute".
- 2) Page 6, lines 139 and 140, please check grammar.
- 3) Page 7, line 182, please consider changing "demonstrated" to "suggest" or "indicate" or similar.
- 4) Page 9, lines 153 and 154, as the BLISS experiments are correlative and did not formally "test whether the genomic instability that we observed upon loss of DDX41 derives from R-loop-associated DSBs", it would be better to rephrase this sentence. In the remaining part of the paragraph, the authors correctly use the terms "correlate" and "coincide" to describe the data.
- 5) In the discussion, a few sentences on the possibility of genomic instability induced by DDX41 loss or mutation that is not causally linked to R-loop formation should be added. On one hand, almost half of the DSBs detected by BLISS at promoters did not display R-loops, and DSB gains were also detected outside promoter regions. On the other hand, several studies reported R-loop induction not as cause, but as consequence of replication stress and DSB formation. Such alternative scenarios could be discussed in more depth.
- 6) Molecular weight markers should be added to Western blot data and scale bars should be added to micrographs.
- 7) In Figure 3d, please check and correct the y-axis.

Reviewer #3:

Remarks to the Author:

The authors have addressed my main concerns in a satisfactory manner

Point-by-point response to the Reviewers' comments

Mosler et al., "R-loop proximity proteomics identifies a role of DDX41 in transcription-associated genomic instability"

REVIEWER COMMENTS

Reviewer #2 (Remarks to the Author):

The authors significantly revised and improved the manuscript. The newly added DSB mapping by BLISS corroborates the conclusion on genomic instability in DDX41-deficient cells and links it to promoter regions of active genes, where also R-loops are found. The data and analyses are overall of high quality. While the new analyses strengthen the correlation between R-loops and DSBs, about half of the promoters with DSBs after DDX41 loss do not seem to display R-loops, and, more importantly, causality between elevated R-loops and DSB induction is not mandatory from the correlative analyses. This should be reflected a bit better in the manuscript text (see below).

Specific recommendations:

- 1) In the abstract, line 37, please consider including "may" before "contribute".

We have added "may" in this sentence.

- 2) Page 6, lines 139 and 140, please check grammar.

Thank you. We corrected this sentence.

- 3) Page 7, line 182, please consider changing "demonstrated" to "suggest" or "indicate" or similar.

We changed the word "demonstrated" to "indicated".

- 4) Page 9, lines 153 and 154, as the BLISS experiments are correlative and did not formally "test whether the genomic instability that we observed upon loss of DDX41 derives from R-loop-associated DSBs", it would be better to rephrase this sentence. In the remaining part of the paragraph, the authors correctly use the terms "correlate" and "coincide" to describe the data.

We rephrased this sentence: "To investigate whether the genomic instability observed upon loss of DDX41 derives from DSBs and whether the sites of DSBs coincide with sites of R-loop accumulation, we performed sBLISS (Break Labeling In Situ and Sequencing) in wild type and DDX41 knockdown HCT116 cells^{48,49}."

- 5) In the discussion, a few sentences on the possibility of genomic instability induced by DDX41 loss or mutation that is not causally linked to R-loop formation should be added. On one hand, almost half of the DSBs detected by BLISS at promoters did not display R-loops, and DSB gains were also detected outside promoter regions. On the other hand, several studies reported R-loop induction not as cause, but as consequence of replication stress and DSB formation. Such alternative scenarios could be discussed in more depth.

We have added the following sentences in the discussion: "It is plausible that R-loop accumulation upon DDX41 loss leads to DSBs by interfering with replication and/or transcription machinery, which might explain why not all sites with R-loop accumulation display increased DNA fragility. It remains to be investigated by which mechanisms and under which conditions dysregulated R-loops are processed into DSBs. We also found sites of increased DNA fragility upon DDX41 loss that did not display R-loop accumulation and hence do not exclude a possibility that DDX41 safeguards actively transcribed genes through additional mechanisms."

- 6) Molecular weight markers should be added to Western blot data and scale bars should be added to micrographs.

We have added molecular weight markers to all western blots and scale bars to all micrographs.

7) In Figure 3d, please check and correct the y-axis.

The y-axis in Figure 3d was corrected. Thank you for pointing this out.